# Global potential of sustainable single-cell protein based on variable renewable electricity

Mahdi Fasihi [1] ✉, Fatemeh Jouzi [1], Petri Tervasmäki[2], Pasi Vainikka[2] & Christian Breyer [1]

The environmental impacts of the food system exceed several planetary boundaries, with protein production being a major contributor. Single-Cell Protein (SCP) is a protein-rich microbial biomass that offers a sustainable alternative when derived from renewable energy and sustainable feedstocks. We evaluate the global potential for SCP production utilising electrolytic hydrogen and oxygen, atmospheric carbon dioxide and nitrogen, and hourly-optimised hybrid PV-wind power plants at a 0.45° × 0.45° spatial resolution. We outline a roadmap for industrial-scale production, commencing in 2028, targeting an annual capacity of 30 million tonnes of protein by 2050. Here we show that the cost of renewable electricity-based protein (e-protein) could decline at optimal sites from 5.5–6.1 € kg⁻¹ in 2028 to 4.0–4.5 € kg⁻¹ by 2030, and further to 2.1–2.3 € kg⁻¹ by 2050. Consequently, e-protein production can mostly decouple protein supply from water and arable land constraints, substantially mitigating the environmental impacts of food production.

Our planet has already exceeded both the safe and just Earth system boundaries[1,2] as well as the planetary boundaries[3–5] for biodiversity loss (biosphere integrity), biogeochemical flows (phosphorus and nitrogen), land-system change, freshwater resources (surface and groundwater), and climate change. The global food system is the primary driver of the first four transgressions and a significant contributor to the last one (climate change), accounting for over a quarter of global anthropogenic greenhouse gas (GHG) emissions[6].

The environmental impacts of the food system are mainly linked to livestock production. Livestock, however, contributes only 25% of global protein and 18% of calorie intake[7]. Due to population growth and rising average incomes, the demand for livestock products is projected to increase by 70% between 2010 and 2050. The environmental impacts of the food system are projected to increase by 50–90% during this period[8,9]. Such growth rates could further push planetary boundaries from uncertainty (increasing risk) to a high-risk zone and then to a tipping point[1,2,4,10]. Thus, scientists are calling for a full transformation of the global food system to a sustainable one to be in line with the United Nations Sustainable Development Goals and the Paris Agreement[10–12]. The call for a transformation of the global food

system highlights the need for alternative resources and technologies to ensure a sustainable and affordable food and feed supply. Achieving this goal requires sustainable supplies of protein, a key component in food, with its availability, quality, and environmental impacts being critical indicators of food security[13].

One of the most effective conventional approaches to mitigate the food system's GHG emissions is switching to a plant-based diet[14]. However, such a switch is neither enough for a fully sustainable food system at a scale required for the growing population[8,15] nor realistically achievable globally by 2050. Other conventional approaches, though less effective, include reducing food supply chain losses[16] and switching from ruminants' products to less-emitting livestock such as poultry[9,17]. More recent and innovative approaches to mitigate food systems' environmental impacts include vertical farming[18], farming insects as food[19], lab-grown meat[20], and microbial proteins[21].

The main types of microbial proteins include algae, yeast, fungi, and bacterial protein[22]. Microbial proteins are generally referred to as microbial biomass or single-cell protein (SCP), even though some may be multicellular[21,23]. The dry mass of microbial biomasses contains 30% to 80% protein, with the rest being edible lipids, carbohydrates, and

[1]LUT University, Lappeenranta, Finland. [2]Solar Foods Oyj, Vantaa, Finland. ✉e-mail: mahdi.fasihi@lut.fi

micronutrients[23]. Among microbial cells, bacterial cells have the fastest doubling time[24]. Microorganisms can be classified according to their energy conversion pathways. Autotrophic microorganisms can directly use carbon dioxide ($CO_2$) as a source of carbon, whereas heterotrophic microorganisms can only use organic carbon for assimilation, growth, and division[25]. Based on the energy source for $CO_2$ assimilation, autotrophic microorganisms are divided into photoautotrophic and chemoautotrophic microorganisms. Photoautotrophic microorganisms use light energy for photosynthesis, whereas chemoautotrophic microorganisms use chemical energy for $CO_2$ assimilation through chemosynthesis[16]. Hydrogen ($H_2$), ammonia ($NH_3$), hydrogen sulphide gas ($H_2S$), ferrous iron ($Fe^{2+}$), and elemental sulphur (S) are among the substances used as the source of energy by chemoautotrophic microorganisms for $CO_2$ assimilation[16].

SCP is not a fully new concept. During World War I, Germany produced yeast biomass on a large scale as a substitute for halted protein feed import[26]. Decades ago, $CO_2$-based microbial biomass production was considered a circular food system in space travel[27]. In the 1970s, Pruteen, a bacterial SCP from methanol, was developed and used as feed[28], but was later discontinued for financial reasons[23]. Pekilo®, a mycoprotein from side streams of the pulp and paper industry, was commercially available in the Finnish feed market from 1974 to 1989. Pekilo® is currently under improvement for reproduction as feed and food[29]. The fungus-based meat substitute, Quorn™, has been commercially available since 1985[30]. FeedKind®, Spirulina, and UniProtein® are other examples of microbial protein-based products on the market[31]. However, access to low-cost, sustainable growth substrates has limited the use of such microbial biomass-based products.

The declining cost of renewable sources of electricity (mainly solar PV and wind power) and water electrolysers have made the production of low-cost green $H_2$ and oxygen ($O_2$) feasible[32]. Moreover, direct air capture (DAC) of $CO_2$ is emerging as an affordable technology for sustainable $CO_2$[33]. Such advancements provide a proper foundation for the development of industrial-scale bacterial SCP plants based on $H_2$-oxidising chemoautotrophic microorganisms using green $H_2$ as the energy source for the assimilation of atmospheric $CO_2$ as the main substrate.

Figure 1 illustrates the schematic of the baseload Power-to-SCP production chain based on a configuration of Solar Foods Oyj[34], as described in the figure caption. The hourly flow rates shown represent the expected values for a full-scale industrial plant in 2030. The energy and mass balance of a baseload system beyond 2030 would differ based on overall advancement in sub-units and projected improvement in bioreactor productivity, as detailed in Supplementary Table 21. Since electricity serves as the primary source of energy and enabler of the feedstocks, we refer to the final product shortly as e-SCP.

e-SCP makes it possible to decouple food production from conventional agriculture, thereby substantially reducing its freshwater and arable land use, along with other environmental impacts[31,38]. The closed system of e-SCP production decouples food production from seasonality of conventional agriculture and impacts of climate change, such as droughts. e-SCP production maximises nutrients utilisation efficiency (substantially limiting nitrogen and phosphorus emissions) and eliminates the need for herbicides or pesticides[22]. e-SCP could also reduce the GHG emissions of food production if supplied with sustainable electricity. The production volume, cost, and emissions of e-SCP are

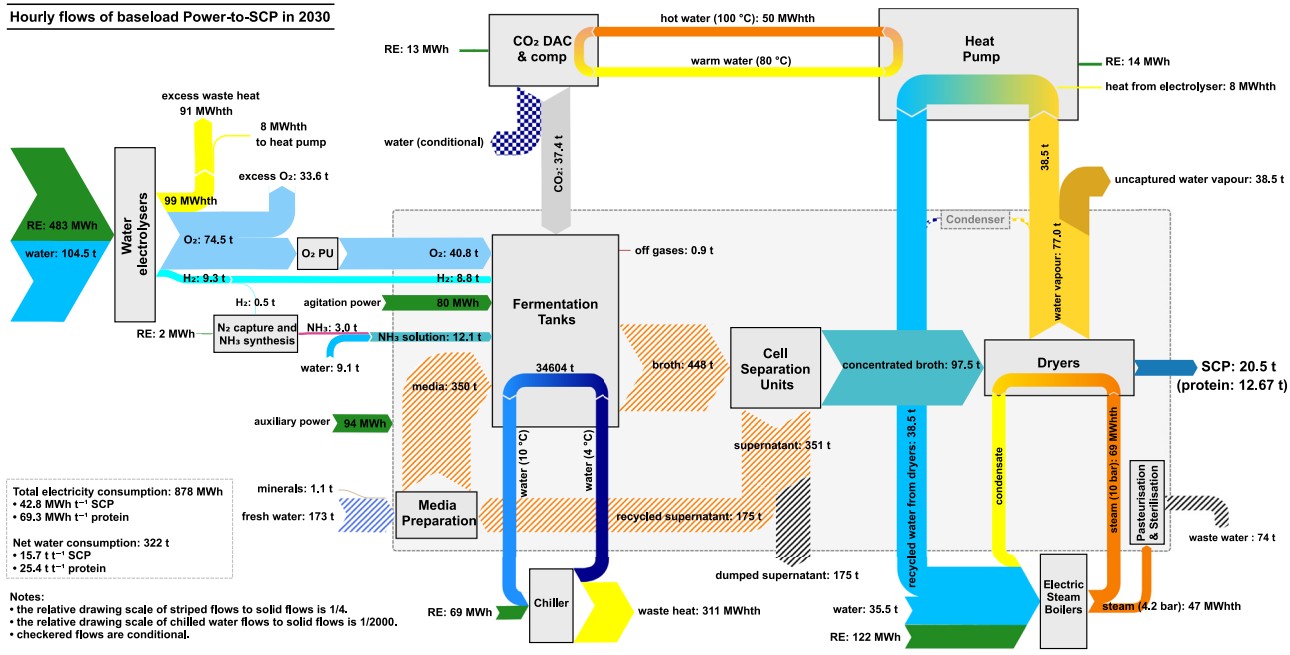

**Fig. 1 | Power-to-SCP chain for a proposed full-scale e-SCP plant in 2030 for baseload operation.** The single-cell protein (SCP) production occurs in 120 fermentation tanks (each 200 m³) by $CO_2$ assimilation of $H_2$-oxidising chemoautotrophic microorganisms at 30 °C. The process requires carbon dioxide ($CO_2$), oxygen ($O_2$), hydrogen ($H_2$), ammonia ($NH_3$) solution, and water, as well as minerals[39] supplied through filter sterilisation. $H_2$ and $O_2$ are supplied by water electrolysers, and the required $CO_2$ is captured by direct air capture (DAC) units. Ammonia is produced onsite via a small-scale Power-to-Ammonia plant. The fermentation process is exothermic and the temperature of the solution in the fermentation tanks is stabilised by a water chiller system. The broth (a liquid medium containing water, unused minerals, and microbial cells referred to as cell dry weight (CDW)) is then pasteurised with low pressure steam and guided to the cell separation units where 97.5% of CDW is separated in the form of concentrated broth with 200 $g_{CDW}$ L$^{-1}$ density. Of the remaining supernatant, mainly water, half is recycled to the media preparation unit and the other half is discarded as wastewater. Then, the concentrated broth is sent to the drum dryer units where most of the remaining water is evaporated via high pressure steam, provided by electric steam boilers. The product SCP is 95% dry matter and 5% moisture. The moisture content is over 96% water and less than 4% non-volatile salts from unreacted minerals in the medium. The dry matter is 65% protein. The waste heat from the electrolysers and the evaporated water from the dryer units are partly retrieved and used as heat sources for heat pumps to provide heat at the desired temperature for the DAC units. The rest of water vapour escapes the system. Additional power is required for agitation in fermentation tanks and auxiliary use in the chain. RE renewable electricity, PU process unit, comp compressor.

largely affected by the availability and cost of renewable electricity, $CO_2$ DAC, water electrolysers, and other energy conversion technologies. These technologies are traditionally linked to the defossilisation of the energy-industry system[35]. Thus, e-SCP could transform the traditional food production space into an energy system space.

Recent scientific publications reflect the renewed interest in SCP production. On a broader range, Ritala et al.[23] reviewed SCP production from various organisms and recent developments in patents and industrial activities in the field. Linder[16] and Nyyssölä et al.[21] reviewed the main options for SCP production from microorganisms and discussed the case for necessity and space for edible microorganisms. Sillman et al.[36] narrowed down the research to water and land use of SCP production via $H_2$-oxidising bacteria, $CO_2$ DAC, and renewable electricity in comparison to those of soybean production. Givirovskiy et al.[37] focused on the impact of in-situ water electrolyser in an electro bioreactor to improve the productivity of $H_2$-oxidising bacterial SCP. Later, Ruuskanen et al.[38] reported on the performance of a pilot-scale project of such a plant. The sustainability aspects of the Power-to-Protein concept were studied through life-cycle assessments by Sillman et al.[31] and Järviö et al.[39]. Pikaar et al.[22] represent the earliest identified peer-reviewed article addressing the economics of SCP production from $H_2$ and point source $CO_2$. Subsequent studies by Nappa et al.[40], García Martínez et al.[41], Leger et al.[42], and Jean and Brown[43] provided more comprehensive case studies on techno-economics of SCP production from electrolytic $H_2$ and $O_2$, as well as from point source or atmospheric $CO_2$. An overview on these articles is provided in Supplementary Note 3, and their key assumptions and results are discussed and compared with our results in the Results section.

The existing literature on the techno-economic assessment of e-SCP production is limited in terms of quantity, geographic scope, modelling approaches, and the originality of data. We identified only five scientific articles addressing the topic, often using predefined fixed prices for baseload electricity and feedstocks supply. Industrial or proprietary data from technology developers regarding the techno-economics of SCP core plants utilising hydrogen-oxidising bacteria (HOB) have not been publicly available. As a result, existing studies primarily rely on assumptions or cost data derived from a methane-based SCP plant design[44]. In this study, we utilise and publicly disclose original techno-economic data from an HOB technology developer with an operational demonstration plant[45]. Additionally, we provide a roadmap for the potential cost trajectory of e-SCP plants over time and with cumulative installed capacity.

The microorganism growth in the e-SCP production process can withstand hours of substrate shortage[46], enhancing its compatibility with variable renewable electricity sources. However, as demonstrated by Nappa et al.[40], a low utilisation rate of the Power-to-SCP production chain could substantially increase SCP production costs. Conversely, the extensive use of power-balancing technologies to ensure a stable baseload electricity supply could substantially raise electricity cost[32], further escalating the cost of e-SCP. In certain regions, a hybrid solar and wind power supply may reduce the seasonality of electricity generation and the need for balancing technologies. Therefore, a cost-optimal solution likely involves a hybrid power supply, flexible operation of subunits, and the integration of various energy and feedstock balancing technologies (see Fig. 2).

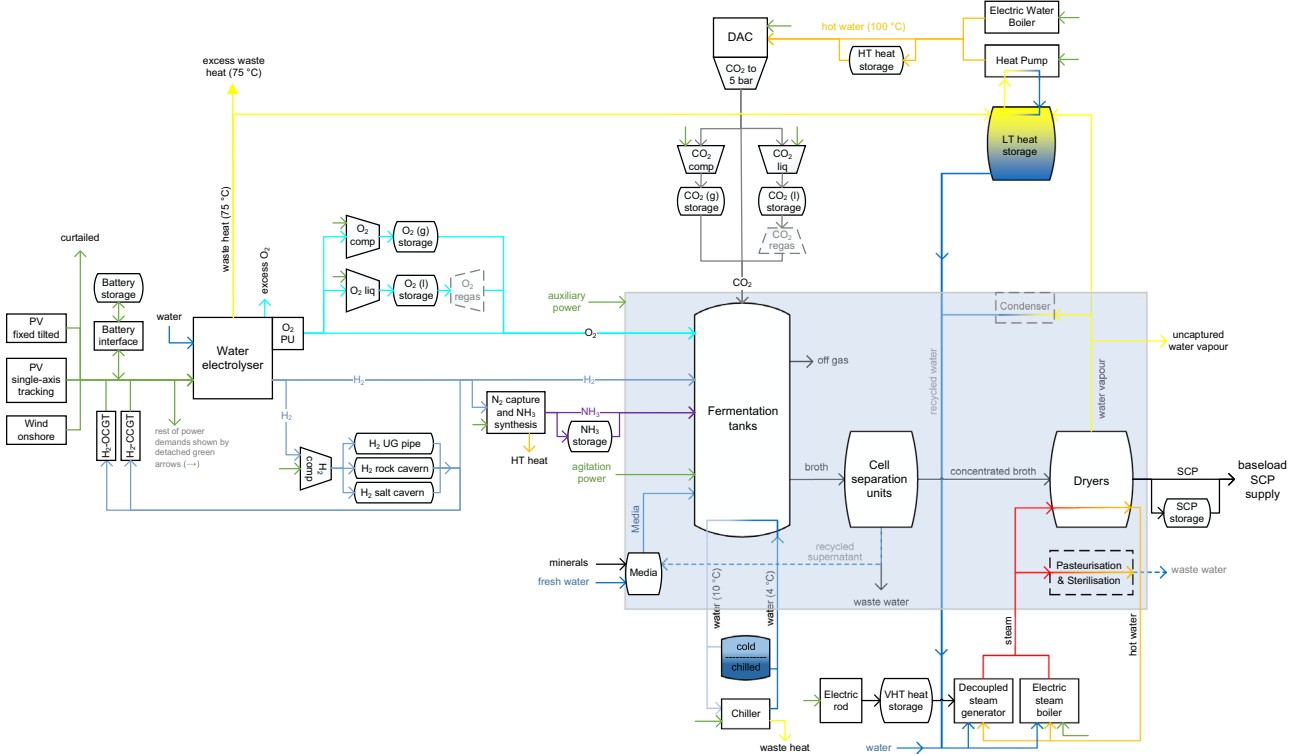

**Fig. 2 | Semi-flexible onsite Power-to-SCP model configuration.** The model includes optimally fixed tilted photovoltaics (PV), single-axis tracking PV, and wind power as the sources of hourly feed-in electricity. Lithium-ion battery and $H_2$-fuelled open cycle and combined cycle gas turbines ($H_2$-OCGT, $H_2$-CCGT) act as daily and seasonal power balancing technologies. The cluster of alkaline water electrolysers provides full range hourly flexibility. Prior to utilisation, the produced $H_2$ could be balanced via the $H_2$ compressor and storage system, namely underground pipes, man-made salt caverns, and lined rock caverns, depending on feasibility according to onsite geological formations. The byproduct $O_2$ of electrolysers is partly captured by $O_2$ processing unit (PU). The system includes both gaseous and liquid storage systems for $O_2$ and $CO_2$ balancing. In addition to electric steam boilers, a second power-to-steam system[75] is included to decouple steam generation from power consumption using very high temperature (VHT) heat storage. Storage units are also considered for balancing ammonia, chilled water, SCP, as well as low temperature (LT) and high temperature (HT) water supply and demand. comp compressor, liq liquefaction, regas regasification, UG underground.

We could not identify any literature modelling large-scale e-SCP production based on variable solar and wind power supply, incorporating the impact of balancing technologies in a techno-economic assessment for any location worldwide. Consequently, the gross energy demand of e-SCP, its land use, and its global production potential have remained understudied. In this research, we address these gaps by modelling a semi-flexible e-SCP plant coupled with energy and feedstock balancing technologies to identify the cost-optimal system configuration within the technical constraints and hourly availability of potential power resources. The model configuration is illustrated in Fig. 2 and further explained in the Methods section. We assess the global production cost of e-SCP through hourly resolved cost-optimised Power-to-SCP systems with a 0.45° × 0.45° spatial resolution for the period from 2028 to 2050. We also present the corresponding technology mix, including solar photovoltaics (PV) and wind power plants, DAC units, and SCP plant installations. Additionally, we provide regional and global theoretical SCP production volumes based on the applied technologies and their land requirements.

## Results

### e-SCP core plant development

The capital expenditures (capex) of the first e-SCP core plant, with a nominal capacity of 16.42 kilo tonnes of SCP per year ($kt_{SCP}$ $a^{-1}$) in 2028, are assessed at 14,567 euros per tonne of protein per year ($€ t^{-1}_{protein}$ a). Our results, as illustrated in Fig. 3a, indicate that the capex of a full-scale e-SCP core plant, with a nominal capacity of 164.2 $kt_{SCP}$ $a^{-1}$ (101.4 $kt_{protein}$ $a^{-1}$), could decline to 8649 $€ t^{-1}_{protein}$ a in 2030, driven by economies of scale. These capex rates are approximately 2–5 times higher than those reported in most earlier studies[22,41–43], as shown in Table 1 and illustrated in Supplementary Fig. 2. Nappa et al.[40] represent the only study with comparable overall capex for the core e-SCP plant in 2028. However, Solar Foods has already achieved volumetric productivity of fermentation tanks that is 7.7 times higher than the value considered in the Nappa et al.[40] system. As a result, for equal production capacities, the system modelled by Nappa et al.[40] would require fermentation tanks with 7.7 times the volume. This suggests that the volumetric cost of fermentation tanks in Nappa et al.[40] is considerably lower than the costs considered in this study.

Beyond 2030, the capex declines by each doubling of the historical cumulative installed capacity (HCIC) based on the considered learning rates (Fig. 3b). In the reference scenario, an S-curve trajectory to achieve a nominal production capacity of 150 million tonne SCP per year ($Mt_{SCP}$ $a^{-1}$), equivalent to ~92.6 $Mt_{protein}$ $a^{-1}$, by 2070 and a learning rate of 10% are considered. The results for the reference scenario show that the capex of the e-SCP core plant could decline to 6175, 3702, and 3041 $€ t^{-1}_{protein}$ a in 2035, 2050, and 2070, respectively (Fig. 3a). In the advanced scenario (doubling the 2070 target e-SCP operational capacity, coupled with a 15% learning rate), the capex of the core e-SCP plant would further decline to 4371, 1985, and 1464 $k€ t^{-1}_{protein}$ a in 2035, 2050, and 2070, respectively (Fig. 3a). For reference, the targeted nominal production capacity of 29.6 $Mt_{protein}$ $a^{-1}$ by 2050 is equivalent to 9.9% and 1.7% of projected food and feed protein demand by 2050, respectively (see Methods for more information).

### Technology mix and cost of electricity supply

The production cost and volume of e-SCP are directly affected by the regional potential (generation cost and volume) of renewable electricity. Our results, as illustrated in Fig. 4a, show that in 2030, wind power will be the dominating electricity source for a cost-optimised PV-wind Power-to-SCP plant in Patagonia, northern Europe, and the UK. Conversely, solar PV would be the main electricity source in the western US, Mexico, Brazil, southern Europe, the whole of Africa, Middle East, Australia, and Asia. By 2050, the PV-dominated regions expand to most of the US and more regions in Patagonia and southern Europe (Fig. 4b). In 2030, the least-cost electricity generation sites, such as the Atacama Desert, Africa, Middle East, and northern Australia, could generate electricity for 15–19 euros per megawatt-hour ($€ MWh^{-1}$), which are all PV-dominated regions (Fig. 4c). By 2050, the cost of electricity generation in least cost sites declines to 8–10 $€ MWh^{-1}$ (Fig. 4d). However, the cost of electricity supply to e-SCP plants is relatively higher at 20–24 $€ MWh^{-1}$ in 2030 (Fig. 4e) and 12–14 $€ MWh^{-1}$ in 2050 (Fig. 4f), resulting from the additional cost of power balancing technologies and curtailment. For comparison, the literature assumes a cost range of 9.5–108 $€ MWh^{-1}$ for baseload Power-to-SCP systems (Table 1), which is generally higher than the cost in this study. It is important to note, however, that the electricity

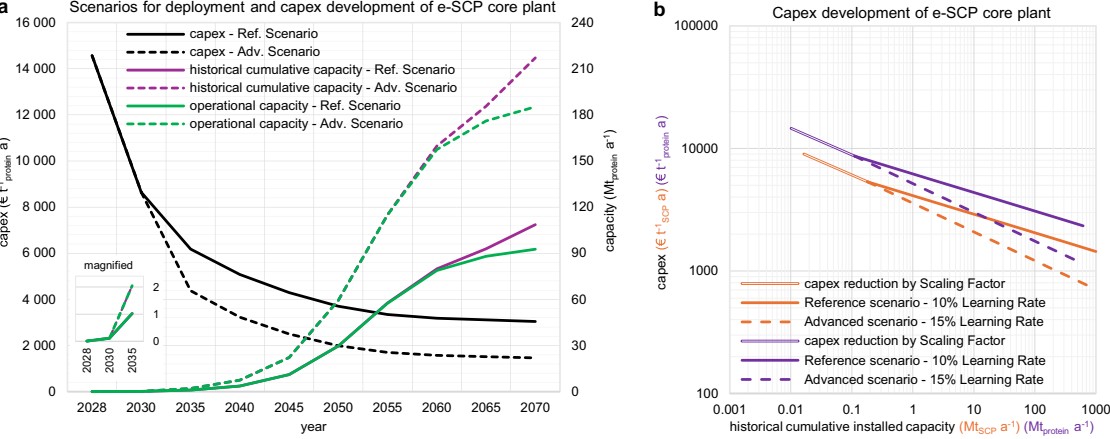

**Fig. 3 | Potential capex decline of e-SCP core plant based on two learning rates and deployment rate scenarios. a** Timewise capex development of e-SCP core plants between 2028 and 2070. The capex in 2028 and 2030 are for the first small-scale and full-scale plants, respectively. The capex decline is achieved by economies of scale for a 10-fold increase in the benchmark plant capacity to 164.2 $kt_{SCP}$ $a^{-1}$. The size of the benchmark full-scale plant remains constant from 2030 to 2070, and the capex decline is achieved by the impact of the learning curve as the historical cumulative installed capacity increases. The historical cumulative installed capacity exceeds the operational capacity beyond 2060, which is due to the additional capacity required for the replacement of retired plants at the end of their lifetime. The e-SCP core plant reinstallations beyond 2060 could be at old sites as brownfield installations, which could further lower the capex. See Supplementary Fig. 3 for a comparable diagram based on total SCP content. **b** Capex development of e-SCP core plants as a function of historical cumulative installed capacity and two scenarios (reference and advanced) for the learning rate. Source data are provided as a Source Data file. Additional data on the development of operational expenditures and newly installed capacity at each timestep are provided in Supplementary Tables 5 and 6. SCP single-cell protein, Ref. reference, Adv. advanced.

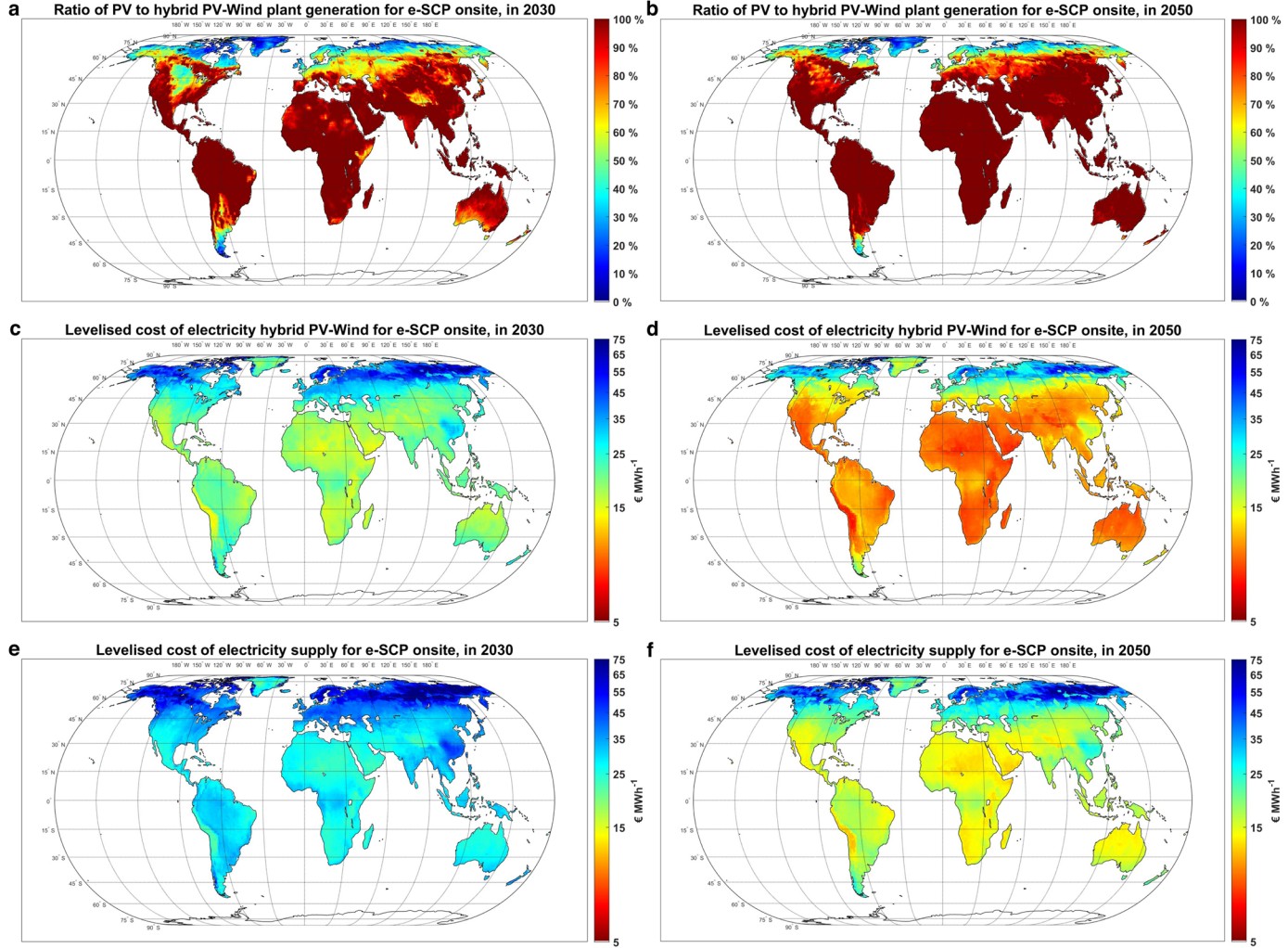

**Fig. 4 | Technology mix and cost of electricity supply in 2030 and 2050.** The years 2030 and 2050 represent the timeline for the first full-scale e-SCP plant, and a well-developed market, respectively. **a, b** Optimal ratio of PV to hybrid PV-wind plant generation for Power-to-SCP plant. **c, d** Levelised cost of electricity generation. **e, f** Levelised cost of electricity supply. SCP single-cell protein. Source data are provided as a Source Data file.

generation and supply in our results are part-load (Supplementary Figs. 7 and 8) and based on projected cost reduction of solar PV and wind power, as well as power balancing technologies. Baseload electricity supply, particularly in regions with pronounced seasonality in solar and wind energy availability, would incur a considerably higher cost[32]. This scenario is avoided by the optimisation model, which aims to minimise the cost of the final product.

### Levelised cost of e-protein

We evaluated the production cost of e-SCP, and consequently, the cost of its protein content (e-protein) from 2028 to 2050, with global protein cost maps for 2028, 2030, 2035 and 2050 illustrated in Fig. 5. Our results indicate that, in 2028, the least-cost e-protein could be produced at 5500–6100 euros per tonne protein (€ t$^{-1}_{protein}$), or 3575–3965 € t$^{-1}_{CDW}$, in regions with the lowest electricity supply costs, such as Patagonia, Northern Australia, the Atacama Desert, Tibet, and the Horn of Africa (Fig. 5a). In contrast, the production cost of e-protein under Nordic conditions in Finland would exceed 7300 € t$^{-1}_{protein}$ (4745 € t$^{-1}_{CDW}$).

In comparison, the literature reports a CDW production cost of 1217–5611 € t$^{-1}_{CDW}$ from baseload Power-to-SCP plants based on various techno-economic and financing assumptions, as shown in Table 1. Recalculating the CDW cost in the literature based on the weighted average cost of capital (WACC) and plant lifetime used in this study

results in a production cost range of 1135–5460 € t$^{-1}_{CDW}$ (see Table 1 and Supplementary Fig. 2d). A detailed cost comparison of our results and the literature is provided in the next subsection (Cost distribution and system dynamics at selected sites). Additionally, an extended techno-economic comparison of Solar Foods' system specifications with literature data is presented in the Supplementary Note 3.

By 2030, the cost of e-protein production from the first full-scale plant in low-cost regions will decline to 4000–4500 € t$^{-1}_{protein}$ (2600–2925 € t$^{-1}_{CDW}$) (Fig. 5b). The cost reduction is primarily due to a substantial decrease in the capex of the SCP core plant and other components impacted by economies of scale.

By 2035, the production cost of e-protein is projected to reach 3100–3500 € t$^{-1}_{protein}$ (2015–2275 € t$^{-1}_{CDW}$) at the best sites (Fig. 5c). By 2050, low-cost e-protein could be produced for 2100–2300 € t$^{-1}_{protein}$ (1365–1495 € t$^{-1}_{CDW}$) at the best sites and may become widely available across all continents at a cost of 2300–2500 € t$^{-1}_{protein}$ (1495–1625 € t$^{-1}_{CDW}$) (Fig. 5d). The decline in e-protein production costs beyond 2030 is associated with gradual reductions in cost and efficiency gains within the Power-to-SCP chain. The global e-protein and e-SCP cost maps for all simulated years are available in Supplementary Figs. 4 and 5.

### Cost distribution and system dynamics at selected sites
In this section, we provide the cost distributions of e-protein and more insight into the dynamics of the cost-optimised systems for 100 kt a$^{-1}$

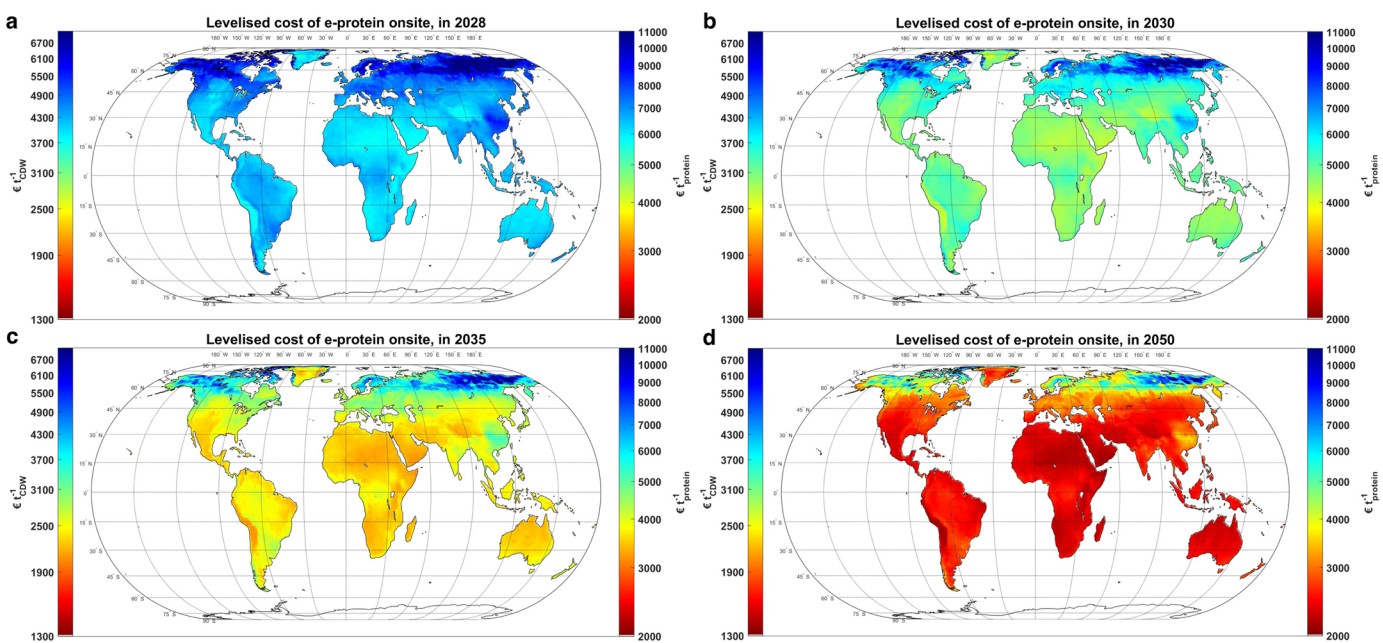

**Fig. 5 | Levelised cost of e-protein. a** in 2028. **b** in 2030. **c** in 2035. **d** in 2050. CDW cell dry weight. Source data are provided as a Source Data file.

e-protein (~154 kt a$^{-1}$ CDW) supply at seven selected sites in 2030. The objectives of this section are: firstly, to identify the main contributors to the energy consumption and cost of e-protein; secondly, to provide an insight into how the sub-units operate and interact in the cost-optimised system, such as the choice of competing technologies and their utilisation rate; thirdly, to put the scale of the required components, particularly power and H$_2$ supply units, into perspective. Such insights help to identify the key factors for further improvements and sensitivity analyses. Moreover, such details would expand the basis for a more detailed analysis and debate on the input data and results of this study by readers.

The selected sites represent various climate and energy profiles, including a wind-dominated site (Chilean Patagonia), a mixed PV-wind site (Germany), and PV-dominated sites located in Australia, the Atacama Desert in Chile, USA California, and Oman). Additionally, a site with Nordic climate (Finland) is included. The coordinates of these sites are available in Supplementary Table 13. The underlying detailed results for the analyses in this section are provided in Supplementary Fig. 6 and Supplementary Tables 14–20.

Figure 6a shows that, in 2030, Power-to-SCP has an overall electricity consumption of 70–73 megawatt-hour per tonne protein (MWh t$^{-1}_{protein}$), or 45.5–47.5 MWh t$^{-1}_{CDW}$, at the selected sites based on the applied technologies. For comparison, the highest reported energy consumption for Power-to-SCP in the reviewed literature is 37.8 MWh $^{-1}_{CDW}$[41]. The variation in total energy demands is mainly due to differences in the electrical energy requirements of the core SCP plants (Table 1) and, to some extent, the inclusion of energy requirement for onsite green ammonia production in this study.

Our results show that the main electricity consumers are water electrolyser (54–56%), SCP core plant (19–20%), electric rod for steam generation system (11–13%), and chiller (8%). Low-temperature solid sorbent DAC and its heat suppliers (electric water boiler and heat pump) together compose only 3–4% of the total electricity consumption.

The high energy intensity of the process, together with the high share of electricity consumption by flexible electrolysers and electric rods, make the overall system compatible in regions with low-cost variable electricity. Accordingly, as illustrated in Fig. 6b, among the selected sites, the PV-dominated sites in Australia, Chile, and Oman, as

well as wind-dominated Chilean Patagonia have the lowest e-protein production cost at 4278–4492 € t$^{-1}_{protein}$ (2781–2920 € t$^{-1}_{CDW}$) in 2030. One of the best sites in Finland, however, would have the highest production cost at 5825 € t$^{-1}_{protein}$ (3786 € t$^{-1}_{CDW}$), which is 36% higher than the least cost production in the Atacama Desert in Chile. Below, we provide the cost share of major units for Power-to-SCP systems in 2030.

The electricity supply, encompassing both electricity generation and balancing technologies, represents the largest share of e-protein production cost at 1598–2708 € t$^{-1}_{protein}$ (1039–1760 € t$^{-1}_{CDW}$) or 37–49% (Fig. 6b). This wide cost range is primarily due to the substantial variation in the levelised cost of electricity supply across different sites (Fig. 6e). In addition to the electricity generation cost, the cost of electricity supply is also affected by the curtailment and cost of power balancing technologies, namely batteries and H$_2$-fuelled gas turbines (Fig. 6a and Fig. 6d).

Power balancing technologies are primarily required to supply the SCP core plant with its direct power demand to maintain a minimum operational level of 50% each hour. Theoretically, all other input flows to the SCP core plant could be supplied through balancing options during periods of insufficient power generation. However, additional power balancing capacities could be installed to increase the Full load hours (FLh) of other sub-units, such as the SCP core plant, DAC, and chiller, as part of a cost-optimised solution.

At PV-dominated sites, batteries are the dominating power balancing technology, which reach 18–24% of the nominal capacity of the required PV-wind plant (Fig. 6d). As such, the levelised cost of electricity supply in PV-dominated sites (22.6–26 € MWh$^{-1}$) could be up to 54% higher than their electricity generation cost (14.7–17.8 € MWh$^{-1}$). The role of power balancing technologies is smaller in sites with a high share of wind power, where a minimum level of power generation could be maintained for longer hours.

The energy cost of each electricity-consuming sub-unit does not correlate linearly with its electricity consumption share because the dynamics of electricity consumption are different by sub-units (Supplementary Fig. 8). For example, battery and H$_2$-fuelled gas turbines would be primarily installed to increase the utilisation rate of SCP core plant, chiller, heat pump, ammonia synthesis unit, and DAC units (Fig. 6c and Supplementary Table 17). Thus, the average cost of

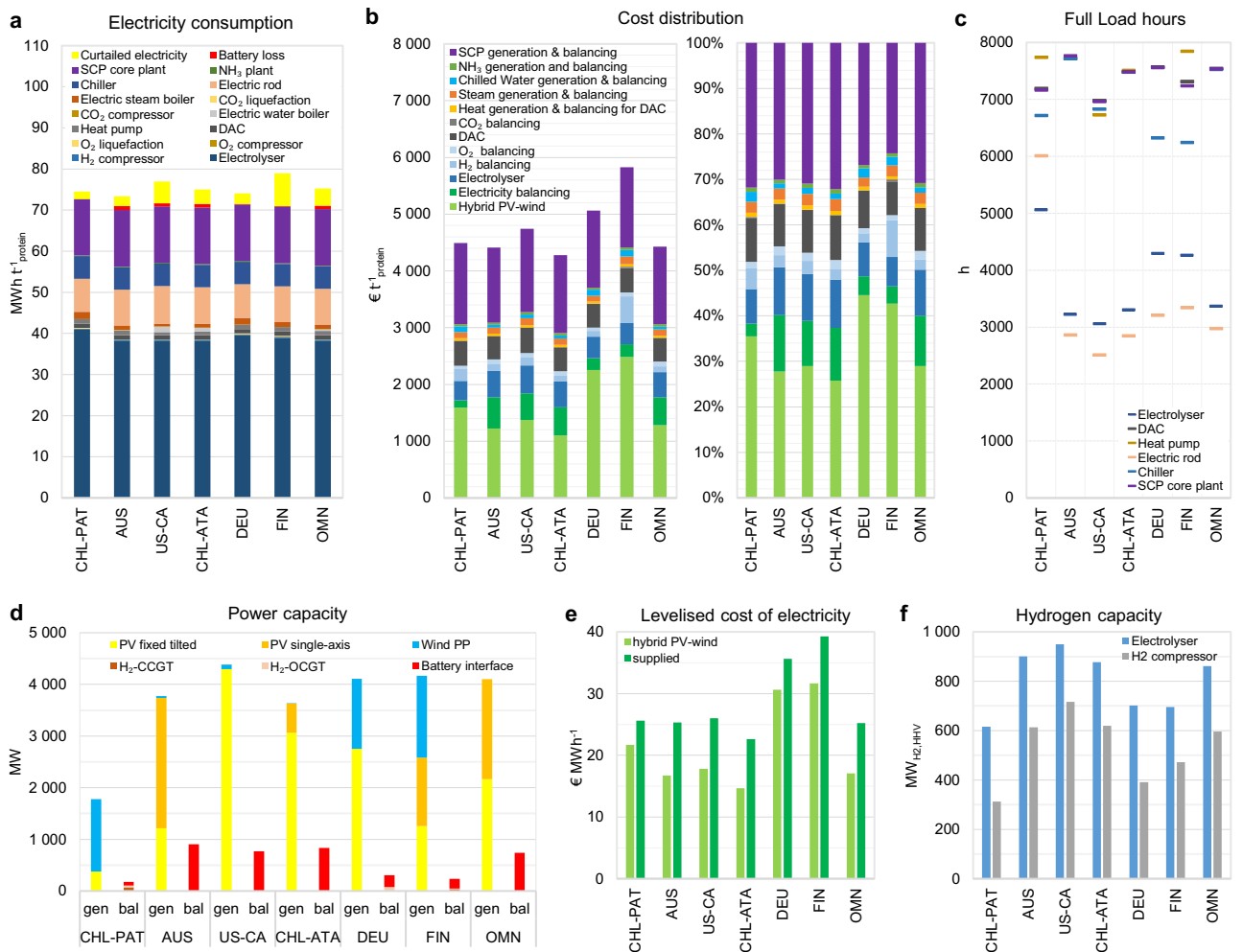

**Fig. 6 | Techno-economics of Power-to-SCP plants for 100 kt a⁻¹ e-protein supply at 7 sites in 2030. a** Electricity consumption. **b** Cost distribution of e-protein. **c** Full Load hours of key components. **d** Installed power and balancing capacities. **e** Levelised cost of electricity. **f** Installed capacities of hydrogen generation and balancing (bottom, right). The selected sites include Chilean Patagonia (CHL-PAT), Australia (AUS), Atacama Desert in Chile (CHL-ATA), USA California

(US-CA), Germany (DEU), Finland (FIN) and Oman (OMN) with exact coordinates in Supplementary Table 13. gen generation technology, bal balancing technology, H₂-OCGT hydrogen-fuelled open cycle gas turbines, H₂-CCGT hydrogen-fuelled combined cycle gas turbines, DAC direct air capture, SCP single-cell protein, HHV higher heating value. Source data are provided as a Source Data file.

electricity supply to sub-units running at higher FLh is more than those running at lower FLh (e.g. electrolyser).

The average cost of electricity supplied at load-following and elevated FLh can be roughly estimated by accounting for the impact of curtailment on total electricity consumption and the cost of power balancing technologies at elevated FLh. Accordingly, the estimated electricity cost at low/high FLh would be 22.2/30.5, 17.5/35.7, 19.4/35, 15.6/32.1, 31.8/41, 35.3/44.6, and 18.3/34.5 € MWh⁻¹ at selected sites in Chilean Patagonia, Australia, Atacama Desert in Chile, California (USA), Germany, Finland, and Oman, respectively. These values can be used for estimation of the energy cost for each Power-to-SCP sub-unit.

Accordingly, the energy cost of H₂ supply (electrolyser and H₂ compressor) would be the highest at 601–1382 € t⁻¹$_{protein}$ (391–898 € t⁻¹$_{CDW}$), followed by the energy cost of SCP plant (including electricity, chilled water, and steam) at 770–1204 € t⁻¹$_{protein}$ (501–783 € t⁻¹$_{CDW}$). The CO₂ supply system (comprising DAC, heat pump, electric water boiler, and CO₂ compression and liquefaction) incurs the third highest energy cost at 66–97 € t⁻¹$_{protein}$ (43–63 € t⁻¹$_{CDW}$). This relatively low energy cost for CO₂ supply is mainly attributed to reduced electricity demand for DAC heating, achieved through the use of waste heat and heat pumps.

The non-energy cost of e-SCP core plants (SCP generation and balancing) has the second highest share by contributing 1327–1469 € t⁻¹$_{protein}$ or 24–32% to the total cost of e-protein in 2030 (Fig. 6b). The non-energy cost of e-SCP core plants differ by less than 11% by site. This small difference occurs because, regardless of potential partial flexibility, the capex-intensive SCP core plants operate at over 7000 FLh in all sites to minimise the installed capacity in the cost-optimised configuration (Fig. 6c).

The non-energy cost of H₂ supply (electrolyser and H₂ balancing), at 479–846 € t⁻¹$_{protein}$ (311–550 € t⁻¹$_{CDW}$) or 9–15%, is the third largest contributor to the total cost of e-protein (Fig. 6b). The lowest non-energy cost of H₂ supply is observed in Germany. Such low cost is achieved through a combination of two factors: firstly, hybrid PV-wind power supply makes it possible to operate electrolysers at relatively high FLh (Fig. 6c), thereby reducing the required installed capacity of electrolyser and H₂ compressor (Fig. 6f); secondly, the site in Germany has access to geological formation for salt cavern as the least-cost H₂ storage option[47] (Supplementary Fig. 15). The installed electrolyser capacity at the west coast of Finland is comparable to those in Germany (Fig. 6f). However, the non-energy cost of H₂ supply in Finland is 78% higher than of those in northern Germany (Fig. 6b). This is because the geological formations at the west coast of Finland are not suitable

for $H_2$ storage; thus, the $H_2$ storage system in Finland is based on more expensive underground pipes (Fig. 6b and Supplementary Table 15).

The total energy and non-energy cost of $H_2$ supply in 2030 could be estimated at 1156–2228 € t$^{-1}$_protein (751–1448 € t$^{-1}$_CDW) or 2416–4656 € t$^{-1}$_H2. This cost range is comparable to those reported in Pikaar et al.[22], García Martínez et al.[41], and Leger et al.[42] (Table 1). In contrast, the $H_2$ supply cost reported by Nappa et al.[40] is considerably higher due to elevated electricity costs and electrolyser capex assumptions. The system analysed by Jean and Brown[43] has the lowest $H_2$ production cost, attributed to the assumption of current baseload wind-based electricity supply at 11.4 USD MWh$^{-1}$ (9.5 € MWh$^{-1}$), which we consider to be unrealistically low without subsidies.

The non-energy cost of $CO_2$ supply (DAC and $CO_2$ balancing) has the fourth largest cost share at 414–457 € t$^{-1}$_protein (269–297 € t$^{-1}$_CDW) or 8–10% of e-protein total production cost (Fig. 6b and Supplementary Table 19). The non-energy cost of $CO_2$ supply remains stable through regions (Fig. 6c), as DAC units run at over 7000 FLh (Fig. 6b) and thus have a similar installed capacity in all selected sites.

As mentioned earlier, the electricity cost of the $CO_2$ supply system is estimated at 66–97 € t$^{-1}$_protein (43–63 € t$^{-1}$_CDW). Consequently, the total cost of $CO_2$ supply would increase to 491–554 € t$^{-1}$_protein (319–360 € t$^{-1}$_CDW). Given a $CO_2$ demand of 2.953 kg kg$^{-1}$_protein, this results in a $CO_2$ supply cost of 166–188 € t$^{-1}$_CO2 by low-temperature solid sorbent DAC in 2030 that benefits from free waste heat in the Power-to-SCP system[40]. For comparison, Leger et al.[42] assumed a $CO_2$ supply cost range of 94–232 USD t$^{-1}$_CO2 based on high-temperature aqueous solution DAC technology. García Martínez[41] is the other study that considers high-temperature DAC in their Power-to-SCP system, for which we reproduced the cost of $CO_2$ supply at 219 € t$^{-1}$_CO2, based on the specified parameters and energy cost assumptions. The remaining reviewed power-to-SCP systems rely on point-source $CO_2$ with a price range of 0–42 € t$^{-1}$_CO2 or 0–98 € t$^{-1}$_CDW, providing a considerable cost reduction for e-SCP production (Table 1). The only exception is Nappa et al.[40], which considers a point-source $CO_2$ purchase price of 40 € t$^{-1}$, along with an additional cost of 138–608 € t$^{-1}$_CO2 for the purification, piping, and storage system, depending on its utilisation rate.

The non-energy cost of $O_2$, heat, steam, chilled water, and ammonia supplies each is negligible. Together, they form the remaining 317–398 € t$^{-1}$_protein (206–259 € t$^{-1}$_CDW) or 8–7% of the total costs or e-protein (Fig. 6b).

## Sensitivity analyses

A series of sensitivity analyses are performed on specifications with considerable impact on the cost of e-SCP in 2030 and 2050 (see Supplementary Figs. 9–14, for illustrations).

The cost evaluations in this study are based on a globally unified WACC of 7%, due to uncertainties with localised WACC by location, time, and investment strategy. The results in Supplementary Fig. 9 show that, in most regions, a ± 2 percentage point deviation in WACC would affect the e-protein production cost by about 400–1100 € t$^{-1}$_protein or 12–15% in 2030. The variation in the impact of WACC values on costs across regions is due to differences in their cost-optimised system configurations. WACC has a greater effect on the capital recovery factor for components with longer lifetimes.

The cost of alkaline water electrolyser widely differs by time, region, and supplier, as shown in Supplementary Table 9. While our Base Cost Scenario (BCS) avoids the two extremes of the range, a High-Cost Scenario (HCS) is considered with 33% higher capex and fixed opex in 2020. Accordingly, following the learning curve path, the capex and fixed opex of the HCS in 2030 and 2050 would be 58% and 62% higher than those in the BCS, respectively. Such elevated electrolyser costs lead to about 150–300 € t$^{-1}$_protein or 3–6% higher e-protein production cost in most regions in 2030 (Supplementary Fig. 10). By 2050, the impact of higher cost electrolyser on e-protein cost will decline to 50–120 € t$^{-1}$_protein or 2–5%. The least absolute

impact is observed in wind-dominated regions wherein a smaller capacity of electrolysers is installed due to higher FLh of wind power and consequently water electrolyser.

DAC plants are at the early stages of industrialisation. Thus, any projections of their future specification face a relatively high level of uncertainty. We consider a HCS for low-temperature DAC wherein only half of the projected cost and energy demand reductions in 2020–2030 and 2030–2050 periods are achieved compared to the BCS. Such a scenario results in 94% and 73% higher capex and fixed opex, 56% and 15% higher electricity demand, and 63% and 18% higher heat demand compared to the BCS in 2030 and 2050, respectively. Accordingly, the e-protein production cost increases by 400–500 € t$^{-1}$_protein or 5–10% in different regions in 2030 (Supplementary Fig. 11). By 2050, the cost increase by HCS is about 100–150 € t$^{-1}$_protein or 2–5% compared to the BCS. The cost increase in the Northern regions is relatively higher due to the higher cost of electricity supply.

The sensitivity of e-protein production cost to the cost of solar PV and wind power plants, as well as SCP core plant, is investigated for a generic ±10% change to their capex and fixed opex in 2030 (Supplementary Fig. 12). A 10% change in the solar PV cost would affect the e-protein production cost by up to 200 € t$^{-1}$_protein or 4% in PV-dominated regions (Supplementary Fig. 13). Nevertheless, such impact is smaller in regions with higher PV full load hours (such as Atacama Desert), as less capacity of PV would be required to meet the energy demands. A 10% change in the wind power plant cost would affect the e-protein production cost by about 100–150 € t$^{-1}$_protein or 2% in centrale Europe. Such impact increases to ~300 € t$^{-1}$_protein or 4% in regions beyond 60° latitude. A 10% change in the SCP core plant cost would impact the e-protein production cost by about 120–140 € t$^{-1}$_protein or 2–3% in most regions (Supplementary Fig. 14).

## Required capacities of key technologies for large deployment

According to Fig. 1, the baseload full-scale Power-to-SCP system with 101 kt annual e-protein supply would require 878 MW of baseload power and 368 MW_H2,HHV (483 MW) electrolyser. Our results in Fig. 6d and Supplementary Table 15 show that a cost-optimised semi-flexible Power-to-SCP system for an annual supply of 100 kt e-protein in the seven selected sites would require 1777–4382 MW hybrid PV-wind nameplate power capacity and 616–949 MW_H2,HHV (808–1245 MW) water electrolyser capacity. Accordingly, the cost-optimal installed power plant and electrolyser in the seven selected sites could be 2–5 and 1.7–2.6 times higher than those in a baseload system, respectively.

Herein, we evaluate the required capacities of PV, wind power, battery, electrolyser, and DAC for large-scale production of e-protein globally. The globally required capacities and annual flows of key Power-to-SCP components for the large-scale supply of e-protein could be estimated based on the installed capacities of the optimised Power-to-SCP systems for e-protein supply in selected regions (Supplementary Table 20). Low-cost e-protein sites are mainly in sunny regions. Thus, we consider the average required power capacities in Australia, Chile (Atacama Desert), and Germany as a rough mix of expected climates for Power-to-SCP plants. Consideration of these sites leads to a global resource mix of 89% to 91% PV power capacity (80% to 83% PV electricity) from 2035 to 2050, respectively for mass deployment of e-SCP plants. As such, 1 and 28 Mt a$^{-1}$ of e-protein supply by 2035 and 2050 would require 33.6 and 905 GW of PV, 4.1 and 93 GW of wind power, 30.0 and 720 GWh_cap of batteries, 8.0 and 231 GW_H2,HHV of electrolysers, as well as 3.4 and 96 Mt_CO2 a$^{-1}$ of DAC, respectively (more information in Supplementary Table 20).

Such capacities may be small compared to their respective demand in a sustainable global energy system by 2050[48]. However, considering the minor impact of currently high-cost DAC plants on e-protein cost (Fig. 6c), a potential business case for e-protein could

act as a niche market for industrial-scale deployment of DAC plants in the late 2020s and early 2030s. The emergence of such markets could have a great impact on DAC technology improvement and their cost reduction for massive deployment starting in the late 2030s.

### Energy-area efficiency and global production potential

We reported an electricity consumption of 69.3 MWh $^{-1}_{protein}$ for the baseload Power-to-SCP configuration in 2030 (Fig. 1). However, our results, illustrated in Fig. 7a, show that the electricity generation for the semi-flexible configuration powered by hybrid PV-wind plants is within the range of 73–83 MWh $^{-1}_{protein}$ in 2030 in most parts of the world. The additional electricity generation is associated with curtailment, battery loss, electricity consumption by compressors, and liquefaction plants, as well as potential utilisation of electric water boilers instead of heat pumps in some regions (Fig. 6a). The electricity generation can be over 90 MWh $^{-1}_{protein}$ in Russian Far East (90°–150° E, 50°–65° N), due to a high level of curtailment as part of the cost-optimal solution (Fig. 7a). The electricity generation is distinguishably lower at sites with access to salt cavern $H_2$ storage, such as Northern Germany and Western Iran, compared to the neighbouring regions. Access to cheap $H_2$ storage in these regions enhances peak power shaving by water electrolyser and thus reduces curtailment. By 2050, the electricity consumption of the baseload Power-to-SCP configuration declines by 8.4 MWh $^{-1}_{protein}$ or 12.1% to 60.9 MWh $^{-1}_{protein}$, of which 3.3 MWh $^{-1}_{protein}$ is due to more efficient water electrolyser and DAC. The improvement in the productivity of the bioreactors, on the other hand, reduces the direct

power, steam, and cooling demand of the core e-SCP plant (see Methods, section 'Energy and mass balance of SCP production'). As a result, the total power consumption by the core plant, electric steam boiler, and chiller declines by 5.1 MWh $^{-1}_{protein}$ in 2050. Consequently, the overall electricity generation at the lower end declines to 64–70 MWh $^{-1}_{protein}$ (Fig. 7b). Nevertheless, the energy demand at the higher end remains over 90 MWh $^{-1}_{protein}$, due to higher curtailment enabled by the reduction of electricity generation cost.

The area demand of hybrid PV-wind Power-to-SCP systems is dominated by the area coverage of hybrid PV-wind plants, whereas the area demand of the rest of the system units is marginal. While PV and wind power plants could be co-located for lower area coverage, Fig. 7c, d present the area coverage of the power system for a case where PV and wind power plants are not co-located. The results show that, in 2030, cost-optimised hybrid PV-wind power plants would have an area coverage of 330–1000 square metres per tonne protein per year (m$^2$ t$^{-1}_{protein}$ a) or 215–650 m$^2$ t$^{-1}_{CDW}$ a in PV-dominated regions. The area coverage could increase to 1200–3000 m$^2$ t$^{-1}_{protein}$ a or 780–1950 m$^2$ t$^{-1}_{CDW}$ a in regions with a large share of wind power (Fig. 7c). For comparison, Nappa et al.[40] reported a PV land use of 69.7 m$^2$ t$^{-1}_{CDW}$ a for e-SCP production in Morocco, which is considerably lower than our results. This difference could be be attributed to the relatively higher overall energy demand of 40.7 MWh $^{-1}_{CDW}$ and lower PV installation density of 75–109 megawatt per square kilometre (MW km$^{-2}$) in our model compared to the reported energy demand of 30.7 MWh $^{-1}_{CDW}$ and a PV density of 200 MW km$^{-2}$ (reproduced) in Nappa et al.[40].

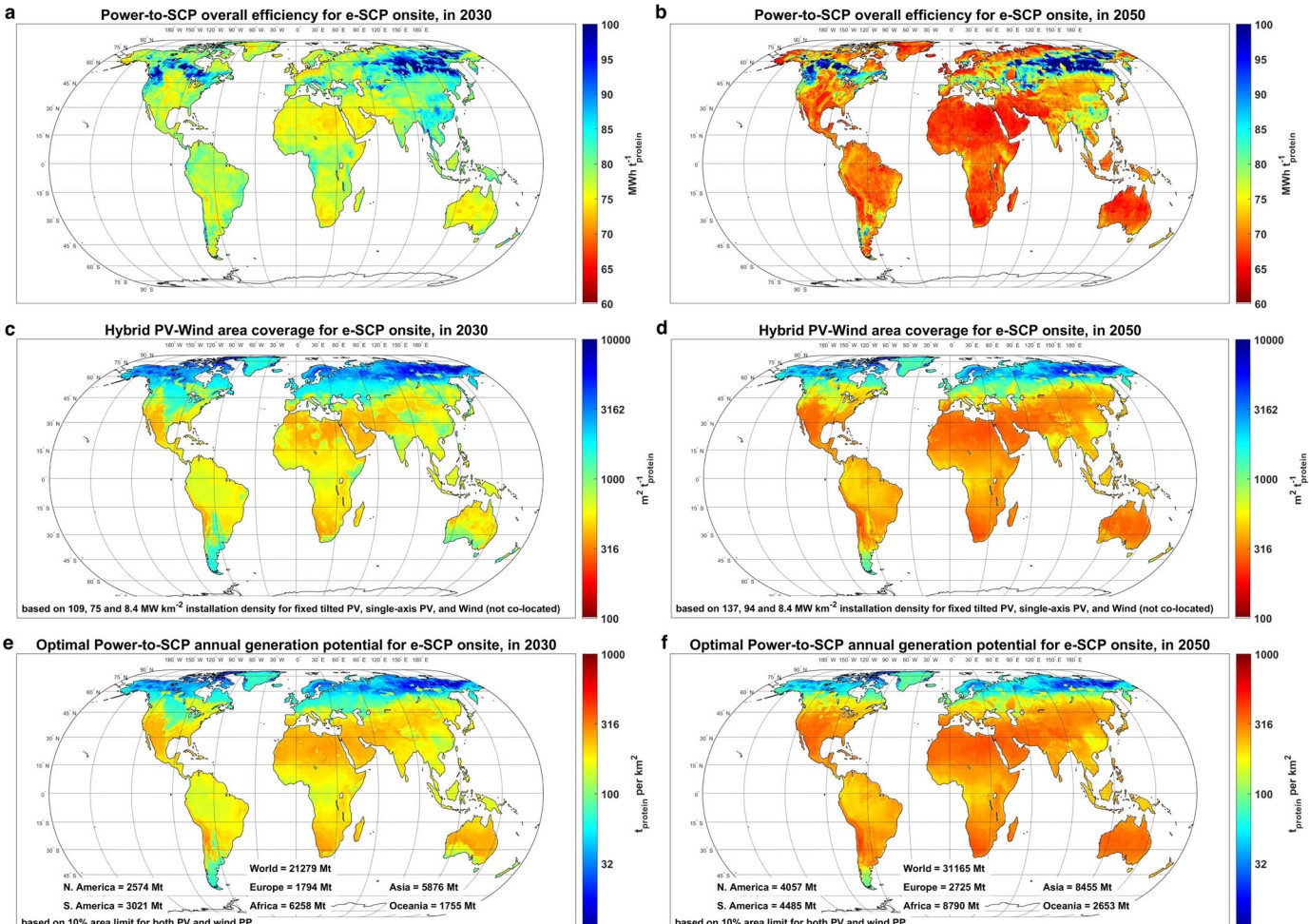

**Fig. 7 | Impact of area demand on global e-protein potential in 2030 and 2050. a, b** Power-to-SCP overall efficiency. **c, d** Hybrid PV-wind area coverage. **e, f** Optimal Power-to-SCP annual generation potential. SCP single-cell protein, PP power plant. Source data are provided as a Source Data file.

## Table 1 | Summary of key techno-economic specifications of SCP core plant from literature and this study

| Annual output t_CDW a^-1 | Capex € t^-1_CDW a | H2 kg kg^-1_CDW | H2 € t^-1 | H2 € t^-1_CDW | O2 kg kg^-1_CDW | O2 € t^-1 | O2 € t^-1_CDW | CO2 kg kg^-1_CDW | CO2 € t^-1 | CO2 € t^-1_CDW | NH3 kg kg^-1_CDW | NH3 € t^-1 | NH3 € t^-1_CDW | electricity MWh t^-1_CDW | electricity € MWh^-1 | electricity € t^-1_CDW | heat MWh t^-1_CDW | heat € MWh^-1 | heat € t^-1_CDW | Opex var € t^-1_CDW | Cost of core plant € t^-1_CDW | Lifetime year | FLh h | WACC % | Cost of product reported € t_CDW^-1 | reprod^a € t_CDW^-1 | recal^b € t_CDW^-1 | Scenario | Reference |
|---|---|---|---|---|---|---|---|---|---|---|---|---|---|---|---|---|---|---|---|---|---|---|---|---|---|---|---|---|---|
| 25,000 | 2500 | 0.56 | 4167 | 2333 | 2.56 | 29 | 75 | 2.32 | 42 | 98 | 0.136 | 549 | 75 | 1.08 | 83 | 90 | 2.92 | 26 | 76 | 27 | 347 | 25 | BL | 5.0 | 3204 | 3121 | 3203 | High-cost H2 | Pikaar et al.[22] |
|  |  |  | 2500 | 1400 |  |  |  |  |  |  |  |  |  |  |  |  |  |  |  |  |  |  |  |  | 2271 | 2188 | 2270 | Average-cost H2 |  |
|  |  |  | 583 | 327 |  |  |  |  |  |  |  |  |  |  |  |  |  |  |  |  |  |  |  |  | 1217 | 1115 | 1197 | Low-cost H2 |  |
| 10,000 | 11,020 | 0.34 | 22,456 | 7635 | 1.24 | 0 | 0 | 1.85 | 648 | 1199 | 0.124 | 695 | 86 | 11.04 | 154 | 1700 | 2.91 | 35 | 102 |  | 12,883 | 20 | 1040 | 10.0 | 24,057 | 23,605 | 18,214 | PV-FIN | Nappa et al.[40] |
|  |  |  | 9725 | 3307 |  |  |  |  | 378 | 699 |  |  |  |  | 51 | 563 |  |  |  |  | 5573 |  | 2409 |  | 10,699 | 10,507 | 8118 | PV-MAR |  |
|  |  |  | 6252 | 2126 |  |  |  |  | 178 | 329 |  |  |  |  | 77 | 850 |  |  |  |  | 1580 |  | BL |  | 5279 | 5128 | 4422 | PVGrid-FIN |  |
|  |  |  | 6872 | 2336 |  |  |  |  | 178 | 329 |  |  |  |  | 88 | 972 |  |  |  |  | 1580 |  | BL |  | 5611 | 5460 | 4755 | PVGrid-MAR |  |
|  |  |  |  |  |  |  |  |  |  |  |  |  |  |  |  |  |  |  |  |  |  |  | BL |  | 2100 |  |  | PVGrid-MAR-best^c |  |
| 100,800 | 2191 | 0.42 | 1757 | 738 | 2.28 | 0 | 0 | 2.41 | 0 | 0 | 0 | - | - | 2.61 | 25 | 65 | 5.59 | 1 | 6 |  | 986 | 20 | BL | 9.3 | (1683) | 1225 | 1135 | Best | Garcia Martinez et al.[41] |
|  |  |  | 6785 | 2850 |  |  |  |  | 219 | 528 |  |  |  |  | 108 | 282 |  | 7.3 | 41 |  | 416 |  |  |  | (5058) | 4117 | 3931 | Worst |  |
| 25,000 | 1667 | 0.43 | 2083 | 896 | - | - | - | 1.76 | 78 | 137 | 0.136 | 375 | 51 | 2.19 | 42 | 92 | 2.27 | 42 | 95 | 28 | 192 | 25 | BL | 3.0 | 1583 | 1491 | 1538 | Low - feed | Leger et al.[42] |
|  |  |  |  |  |  |  |  |  |  |  |  |  |  | 2.77 |  | 116 |  |  |  |  | 230 |  |  |  | 1583 | 1553 | 1610 | Low - food |  |
| 108,000 | 1937 | 0.47 | 3333 | 1567 |  |  |  |  | 193 | 340 |  | 598 | 81 | 4.60 | 83 | 382 |  | 83 | 188 | 49 | 363 |  |  | 8.0 | 2833 | 2970 | 2955 | High - feed |  |
|  |  |  |  |  |  |  |  |  |  |  |  |  |  | 7.91 |  | 657 |  |  |  |  | 508 |  |  |  | 3167 | 3390 | 3369 | High - food |  |
| 45,000 | 3376 | 0.441 | 1675 | 739 | 1.02 | 0 | 0 | 1.84 | 0 | 0 | 0.133 | 458 | 61 | 0 | 9.5 | 0 | 0 | - | 0 |  | 767 | 20 | BL | 14.5 | (1725) | 1567 | 1136 |  | Jean and Brown[43] |
| 15,600 | 9468 | 0.451 | LD | LD | 2.09 | LD | LD | 1.92 | LD | LD | 0.156 | LD | LD | 12.47 | LD | LD | 5.96 | LD | LD | 75 | Depends on FLh | 25 | Semi-flexible | 7.0 | 3400-4580 |  |  | In 2028 | this study |
| 156,000 | 5622 |  |  |  |  |  |  |  |  |  |  |  |  | 12.47 |  |  | 5.96 |  |  | 75 |  |  |  |  | 2470-3600 |  |  | In 2030 |  |
| 218,400 | 2406 |  |  |  |  |  |  |  |  |  |  |  |  | 9.93 |  |  | 5.26 |  |  | 67 |  |  |  |  | 1300-2220 |  |  | In 2050 |  |

BL baseload, crf capital recovery factor, LD location-dependent, NPV net present value method, reprod reproduced, recal recalculated at unified crf, WACC weighted average cost of capital, FLh full load hours, CDW cell dry weight.

Original data in USD was converted to Euro using a long-term USD €^-1 exchange rate of 1.2.

^a The cost of CDW is calculated based on available specifications to reproduce the reported results using the levelised cost method. For references that originally report the minimum selling price (values in parentheses) for a NPV of zero, the WACC value required for the levelised cost method is calculated using the financial assumptions provided in the respective articles.

^b For a better comparison of the results of the literature and this study, the reproduced CDW costs are recalculated based on a unified crf, assuming a 7% WACC and a 25-year SCP plant lifetime. All other specifications remained as stated in the respective references.

^c The 'best' scenario is based on improved productivity, capex, fermentation efficiency, and electricity cost.

In the case of a wind-based system, Jean and Brown[43] reported on a land use of 23.5 m² t⁻¹_CDW a, based on energy demand of 21.2 MWh⁻¹_CDW for H₂ production only and a wind power installation density of 330 MW km⁻², which we find to be infeasible. In contrast, we consider a wind power installation density of 8.4 MW km⁻². Nevertheless, the area impact of PV and wind power plants is about 50%[49] and 1–2%[50] of the stated gross area coverage, respectively, as the space between units could be potentially utilised for other purposes.

By 2050, the area coverage will decline in most regions (Fig. 7d) due to a combination of higher PV panel efficiency and installation power density, lower electricity demand, and higher shares of PV, which has a higher yield per area coverage compared to wind power. For an average area coverage of 400 m² t⁻¹_protein at low-cost sites (Fig. 7d), a supply of 27 and 85 Mt e-protein by 2050 and 2070 (reference scenario) would require ~10,800 and 33,800 km² of land, respectively. For comparison, a 27 Mt e-protein supply in 2050 is equivalent to 9.0% and 1.6% of projected global food protein (300 Mt[51] – Supplementary Fig. 1) and feed protein (1700 Mt[22]) demand in 2050, respectively. To put this into perspective, theoretically supplying 100% of the global protein demand for food and feed in 2050 through e-protein would require 0.54% of the global land area for respective solar electricity generation, which could potentially be on uncultivable land.

On a global scale, we assume an average maximum 10% regional land limit for PV and wind power plant installation each to minimise the potential of land use conflicts and availability for mass deployment of e-SCP plants. As a result, the e-SCP plants could have a theoretical annual production potential of 30–300 t_protein per km² of area in 2030 (Fig. 7e). The high-generation regions are mainly PV-dominated, such as the Atacama Desert, central Australia, west of South Africa, and the Middle East and North Africa region. By 2050, the highest theoretical annual production potential reaches 350–450 t_protein per km² of area (Fig. 7f), as a combination of a more efficient Power-to-SCP chain and higher solar panels installation power density. We observe the highest relative increase in annual e-protein generation potential in Patagonia and Western Europe, with an increase of about 100% from 2030 to 2050, as the share of PV in their cost-optimised system considerably increases over time.

Our results show that, within the defined land use limits, the global theoretical production potential of e-protein in 2050 is about 31 Gt (Fig. 7f). This potential is ~360 times the target for global e-protein supply by 2070 (reference scenario), which enables full flexibility with site selection for e-SCP production at technically feasible and economically suitable regions. This high global potential also makes it likely for most countries or regions in the world to be self-sufficient in generating their e-protein, thus increasing food security and minimising food transportation emissions.

## Discussion

The contribution of alternative proteins to the global food and feed supply depends on their scalability, environmental impacts, safety approvals[52], daily intake limits[22], security of supply, functionality as a food ingredient[53], acceptance by consumers, and market competitiveness. e-SCP has already received approval for use as a food ingredient from the Singapore Food Agency[54] and is seeking approvals in other markets. In this study, we demonstrated that there are no technological obstacles or limitations in land availability and electricity supply in scaling the Power-to-SCP supply chain.

As a food ingredient, e-SCP should compete with soy and pea isolates, which are the main plant-based protein-rich food ingredients with comparable functionality. The recorded regional and global average price of soy isolates were 1.8–8.1 € kg⁻¹_protein[55,56] in 2019–2023 and pea isolate prices were 3.5–9.3 € kg⁻¹_protein in 2020–2023[56], with both stabilising at the higher end since early 2021 (Fig. 8a). Our results showed that e-protein production costs of 5.5–6.0 and 4.0–4.4 € kg⁻¹ are already achievable for the first small-scale and full-scale SCP plants at best sites in 2028 and 2030, respectively. Thus, e-SCP could be a competitive source of protein for food production as early as 2028. As the cost of mass-produced e-protein declines to 2.1–2.3 € kg⁻¹ in 2050, it becomes competitive with low-price soy and pea isolates (Fig. 8a).

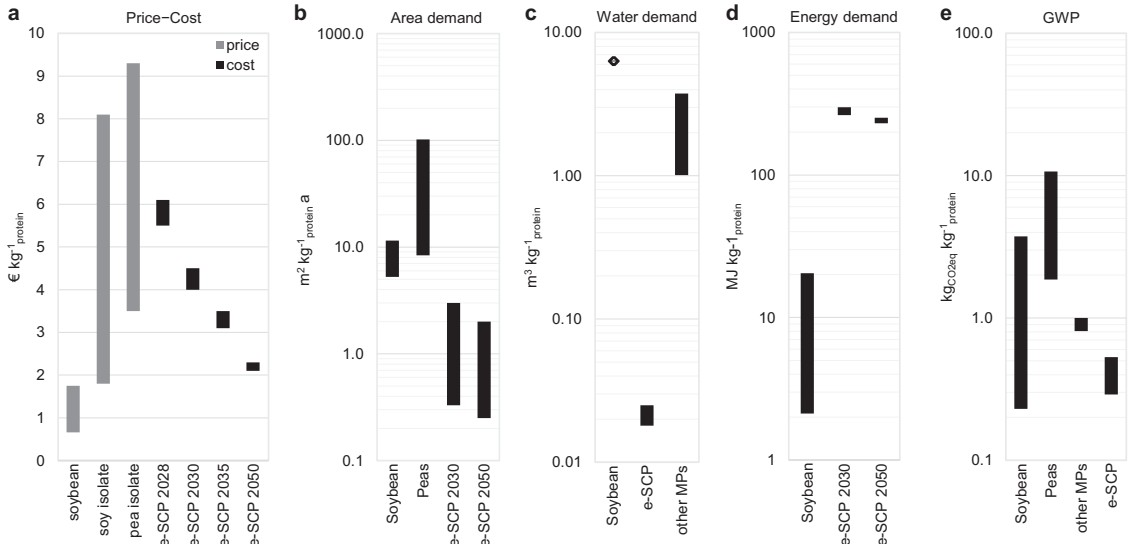

**Fig. 8 | Financial and environmental comparison of protein supply by soybean, soy isolate, pea isolate, and e-SCP. a** Price-cost. The price range of soybean is based on trading prices in countries with an annual soybean trading of over 0.1 Mt in 2010–2021 and a 38% protein content. The protein prices of soy and pea isolates are based on protein content of 85 and 82.5%, respectively. e-SCP cost range is based on production at best sites with 5 Gt_protein a⁻¹ cumulative theoretical production potential. **b** Area demand. The value range for e-SCP represents the area coverage of the required PV-wind power plant and not its direct land use. **c** Water demand. The range for e-SCP represents 2030–2050 values. **d** Energy demand. The energy demand for soybean refers to fossil fuel energy used during pre-farm and on-farm activities including, but not limited to, fertiliser production, infrastructure construction and machinery use[79]. The energy demand range for e-SCP represents electricity generation and not final consumption. **e** Global Warming Potential. Diagrams based on separate data on soybean price[59], soy and pea isolates[57,58], land use[14,31,79], water demand[14,31], energy demand[79], and Global Warming Potential[14,31,39,79,80]. MP microbial protein, GWP global warming potential, e-SCP renewable electricity-based single-cell protein. Source data are provided as a Source Data file.

As a feed, e-SCP should compete with soybean. The annual average price of soybean bulk trading in the last decade was 0.3–0.8 USD kg$^{-1}$[57], equivalent to 0.66–1.75 € kg$^{-1}_{protein}$ (Fig. 8a), for an average protein content of 38%. Thus, with current market conditions and prices, e-protein may not become a competitive feed at scale. However, regardless of the up to 100 times higher energy demand (Fig. 8d), e-SCP, compared to soybean, has about 2–35 times lower land use (Fig. 8b), over 200 times lower water demand (Fig. 8c), and mostly lower global warming potential (Fig. 8e). Thus, new regulations that limit the respective environmental impacts of food and feed supply could increase the competitiveness of e-protein in the market. For example, in 2023, a regulation came into force in the European Union banning some agricultural products and their derivatives (including soy and beef) that are linked to global deforestation and forest degradation from the beginning of 2020[58]. As deforestation is mainly linked to cropland and pasture, the new regulation limits land availability for feed production, thereby could ramp up the use of e-protein as feed. The direct use of SCP as a feed supplement would reduce the GHG emissions of the digestion system of ruminants by reducing their fibre consumption for the same protein intake. Thus, if GHG emission costs be applied to food products, e-protein would become a more attractive choice as a feed for ruminants because it reduces both feed and animal production-related GHG emissions.

The long-term production cost of e-SCP could further decline by lowering the overall energy demand of the process through improved heat integration, a higher productivity rate, and a lower H$_2$/CO$_2$ consumption ratio. Finally, this study was performed for a globally uniform WACC of 7%. A WACC of 5% in applicable regions could reduce the cost of e-protein production by about 15%.

e-Protein could decouple the food supply from the decline in global arable land, increase food sovereignty across the world, keep the global food supply on an affordable level, and help to reduce the pressure on remaining ecosystems by lowering the environmental impacts of the global food supply.

## Methods

Throughout this study, the following naming style is used: The carbon-hydrogen-oxygen-nitrogen content of the desired biomass is called CHON content. The total mass of biomass, including CHON and additional mass originating from the consumed minerals, is defined as CDW. The CDW produced in the fermentation tanks is referred to as CDW$_{gross}$. The mass of CDW leaving the factory is less than CDW$_{gross}$ due to losses in the separation of CDW from the broth in the cell separation unit. The product leaving the factory is referred to as single-cell protein or SCP, consisting of 95% CDW and 5% moisture.

The techno-economic specifications of a potential design for a baseload small industrial-scale SCP plant by Solar Foods in 2028 are utilised as an example of H$_2$-oxidising SCP plants. This design is then modified to align with our system configuration, which is based on the variable-load operation. The techno-economic specifications of large industrial-scale plants beyond 2028 are developed in this study using scaling factors (SF) and learning curves, as these are not provided by Solar Foods. All input and result data are provided in the Source Data file and Supplementary Information.

### Energy and mass balance of SCP production

The microbial strain used in Solar Foods' product is *Xanthobacter* sp. SoF1 as described in Klinzing et al.[59] and Holmström and Pitkänen[60]. The stoichiometry approximation of the CHON composition of the reaction is presented in Eq. (1), which is based on a partial modification of the stoichiometry in Eq. (2) presented in the experimental study by Ishizaki and Tanaka[61]. The CHON composition of the biomass (CH$_{1.7}$O$_{0.37}$N$_{0.21}$) has a molar mass of 22.56 grams per mol (g mol$^{-1}$).

Minerals consumed during the process account for 5% of the CDW, increasing the overall biomass molecular weight to approximately 23.75 g mol$^{-1}$.

$$CO_2(g) + 5.165H_2(g) + 1.5O_2(g) + 0.21NH_3(aq) \rightarrow CH_{1.7}O_{0.37}N_{0.21}(s) + 4.63H_2O(l) \tag{1}$$

$$CO_2(g) + 5.22H_2(g) + 1.52O_2(g) + 0.19NH_3(aq) \rightarrow CH_{1.74}O_{0.46}N_{0.19}(s) + 4.56H_2O(l) \tag{2}$$

Solar Foods employs a continuous closed recirculating process with a 99% feedstocks utilisation rate in the bioreactor. The remaining 1% of the feed gases are vented through exhaust gas management. Of the produced CDW, 97.5% is separated from the broth by the cell separation unit and additional filtration, after which it is sent to the dryer (Fig. 1). The supernatant leaving the cell separation unit contains the remaining 2.5% of the CDW. Half of this supernatant is recycled back to the media preparation unit. However, the CDW in the recycled portion becomes unusable due to sterile filtration of the media with a 0.2 μm filter. The rest of the supernatant is treated as wastewater, resulting in an overall CDW capture of 97.5%. Consequently, the total gaseous feedstocks required to supply 1 kg of CDW are 3.6% higher than the stochiometric mass balance would suggest.

The volumetric productivity of the fermentation tanks is a key factor in SCP production systems. Higher productivity decreases the required size of bioreactor tanks and utilities for the same output capacity, which in return helps reduce the plant's capital investment costs. Productivities as high as 2.6 gram per litre per hour (g$_{CDW}$ L$^{-1}$ h$^{-1}$) have been reported in the literature[62]. Solar Foods has improved the average volumetric productivity of its design from 0.8 g$_{CDW}$ L$^{-1}$ h$^{-1}$ in 2019 to 1.0 g$_{CDW}$ L$^{-1}$ h$^{-1}$ in 2024[63]. For this study, it is assumed that the productivity of future Solar Foods plants will remain at 1.0 g$_{CDW}$ L$^{-1}$ h$^{-1}$ until 2030, after which it gradually increases to 1.4 g$_{CDW}$ L$^{-1}$ h$^{-1}$ by 2050.

The O$_2$/CO$_2$ molar consumption ratio in fermentation is another key factor in SCP production. A lower O$_2$/CO$_2$ ratio also corresponds to a lower H$_2$/CO$_2$ molar consumption ratio and reduced water formation. A lower H$_2$ consumption improves both the economics and overall efficiency of the system. In this study, we retain an O$_2$/CO$_2$ molar ratio of 1.5, and consequently, a H$_2$/CO$_2$ molar ratio of 5.165, from 2028 to 2050. For comparison, Nappa et al.[40] considers a base Power-to-SCP system with a H$_2$-to-Biomass efficiency of 45.5%, corresponding to an O$_2$/CO$_2$ molar ratio of approximately 1. In an improved scenario, Nappa et al.[40] considers an overall efficiency of 55%, which further reduces O$_2$ and H$_2$ requirements.

Heat generation in the fermentation tanks is primarily driven by O$_2$ consumption and the power required for agitation. For a fixed O$_2$/CO$_2$ consumption ratio, the heat generated per kg of CDW remains constant. The total power required for agitation is determined based on the working volume of the tank, at 4 kilowatts per cubic metre (kW m$^{-3}$). At higher productivity rates, more CDW can be produced in the same fermentation tank with the same agitation power, resulting in lower power requirements and reduced demand for chilled water per kilogram of CDW. Similarly, the fixed amount of steam required for periodic sterilisation would translate to less steam consumption per kg of CDW at higher productivity rates. The energy and mass balance for the system at different productivity levels is presented in Table 2. Furthermore, there is potential to enhance the overall energy efficiency of the SCP plant through improved heat integration, though such improvements have not been considered in this study.

### SCP plants deployment and its development

Solar Foods is a start-up company aiming for SCP supply at an industrial scale. The company's pilot unit with a 200 L fermentation tank was

**Table 2 | Techno-economic development of SCP core plants from 2028 to 2050**

| Year | | 2028 | 2030 | 2035 | 2040 | 2045 | 2050 |
|---|---|---|---|---|---|---|---|
| Productivity | $g_{CDW}$ $L^{-1}_{WC}$ $h^{-1}$ | 1.0 | 1.0 | 1.1 | 1.2 | 1.3 | 1.4 |
| $O_2/CO_2$ consumption - molar ratio | - | 1.5 | 1.5 | 1.5 | 1.5 | 1.5 | 1.5 |
| *Benchmark SCP plants* | | | | | | | |
| Total volume of fermentation tanks | $m^3$ | 2400 | 24,000 | 24,000 | 24,000 | 24,000 | 24,000 |
| Working volume of fermentation tanks | $m^3$ | 2000 | 20,000 | 20,000 | 20,000 | 20,000 | 20,000 |
| CDW production - gross (bioreactor output) | $kg_{CDW}$ $h^{-1}$ | 2000 | 20,000 | 22,000 | 24,000 | 26,000 | 28,000 |
| CDW production - net (factory output) | $kg_{CDW}$ $h^{-1}$ | 1950 | 19,500 | 21,450 | 23,400 | 25,350 | 27,300 |
| SCP production - net (factory output, incl. moisture) | $kg_{SCP}$ $h^{-1}$ | 2052.6 | 20,526 | 22,579 | 24,632 | 26,684 | 28,737 |
| *Energy and mass balance* | | | | | | | |
| $H_2$ | $kg$ $kg^{-1}_{SCP}$ | 0.4281 | 0.4281 | 0.4281 | 0.4281 | 0.4281 | 0.4281 |
| $O_2$ | $kg$ $kg^{-1}_{SCP}$ | 1.989 | 1.989 | 1.989 | 1.989 | 1.989 | 1.989 |
| $CO_2$ | $kg$ $kg^{-1}_{SCP}$ | 1.824 | 1.824 | 1.824 | 1.824 | 1.824 | 1.824 |
| $NH_3$ | $kg$ $kg^{-1}_{SCP}$ | 0.148 | 0.148 | 0.148 | 0.148 | 0.148 | 0.148 |
| Minerals[39] | $kg$ $kg^{-1}_{SCP}$ | 0.0543 | 0.0543 | 0.0543 | 0.0543 | 0.0543 | 0543 |
| Power – agitation | $kWh$ $kg^{-1}_{SCP}$ | 3.897 | 3.897 | 3.543 | 3.248 | 2.998 | 2.784 |
| Power – auxiliary | $kWh$ $kg^{-1}_{SCP}$ | 4.577 | 4.577 | 4.266 | 4.007 | 3.788 | 3.600 |
| Steam (10 bar) – drum dryer | $kW_{th}$ $kg^{-1}_{SCP}$ | 3.354 | 3.354 | 3.354 | 3.354 | 3.354 | 3.354 |
| Steam (4 bar) – pasteurisation and sterilisation | $kW_{th}$ $kg^{-1}_{SCP}$ | 2.303 | 2.303 | 2.094 | 1.919 | 1.772 | 1.645 |
| Chilled water utilisation | $kg$ $kg^{-1}_{SCP}$ | 1686 | 1686 | 1635 | 1593 | 1557 | 1 526 |
| Process water [a] | $kg$ $kg^{-1}_{SCP}$ | 10.33 | 10.33 | 9.04 | 7.93 | 6.96 | 6.11 |
| Wastewater (from dryer, steam boiler and cell separation) | $kg$ $kg^{-1}_{SCP}$ | 13.73 | 13.73 | 12.44 | 11.33 | 10.37 | 9.51 |
| Exhaust gas (10% $H_2$, 37% $O_2$, 50% $CO_2$, 3% $H_2O$) | $kg$ $kg^{-1}_{SCP}$ | 0.0425 | 0.0425 | 0.0425 | 0.0425 | 0.0425 | 0.0425 |
| *Deployment and cost development* | | | | | | | |
| Operational capacity | $Mt_{SCP}$ $a^{-1}$ | 0.0164 | 0.181 | 1.66 | 6.0 | 18.0 | 48.0 |
| Historical cumulative capacity | $Mt_{SCP}$ $a^{-1}$ | 0.0164 | 0.181 | 1.66 | 6.0 | 18.0 | 48.0 |
| Capex - 10% LR | € $t^{-1}_{SCP}$ $a$ | 8995 | 5341 | 3813 | 3136 | 2654 | 2286 |
| | € $kg^{-1}_{SCP}$ $h$ | 71,960 | 42,728 | 30,504 | 25,088 | 21,232 | 18,288 |
| Opex $_{fixed}$ | € $t^{-1}_{SCP}$ $a$ | 503 | 261 | 193 | 162 | 140 | 123 |
| Maintenance and insurance | % of capex | 4 | 4 | 4 | 4 | 4 | 4 |
| Personnel - 5% LR | € $t^{-1}_{SCP}$ $a$ | 113.0 | 33.9 | 28.8 | 26.1 | 24.1 | 22.4 |
| Others - 5% LR | € $t^{-1}_{SCP}$ $a$ | 30.6 | 13.3 | 11.3 | 10.2 | 9.4 | 8.8 |
| Opex $_{variable}$ | € $t^{-1}_{SCP}$ | 75 | 75 | 73 | 70 | 69 | 67 |

*CDW* cell dry weight, *SCP* single-cell protein, *WC* working capacity.

[a]For media preparation, steam boiler, and ammonia solution. The water required for $H_2$ production is separately accounted for in variable opex of water electrolyser.

built in 2018. Their demonstration plant (Factory 01) with a 20 $m^3$ fermentation tank and up to 160 tonnes of annual SCP production became operational in April 2024[45]. The company's first commercial-scale plant with a total fermentation working capacity of 2000 $m^3$ is planned for 2028. This capacity has been achieved by the use of 12 fermentation tanks of 200 $m^3$. The difference in the total volume of the fermentation tanks and the working capacity is occupied by gases at the head of each tank. The first full-scale plant with 10 times higher capacity could be expected as soon as 2030. Three engineering consulting firms were contracted by Solar Foods to perform independent techno-economic assessments of a baseload SCP plant with 2000 $m^3$ working capacity under different specifications and configurations. We took the most conservative scenarios from these three techno-economic assessments and further expanded them to match a system operating with variable renewable sources of electricity and feedstocks supply in 2028. Next, we assessed the installation costs of the first full-scale 20,000 $m^3$ plant in 2030 based on SF provided in Supplementary Table 1 via Eq. (3).

$$\text{total capex}_{new} = \text{total capex}_{ref} \left( \frac{\text{capacity}_{new}}{\text{capacity}_{ref}} \right)^{SF} \qquad (3)$$

As more e-SCP plants are built and the HCIC increases, the unit cost of e-SCP core plants is expected to decline. The unit cost of an e-SCP core plant in year 2035 and beyond is evaluated based on standard log-linear learning curve approach via Eq. (4–6), according to Caldera and Breyer[64]. Abbreviation: binary exponential expression of the progress ratio (b).

$$\text{unit capex}_{new} = \text{unit capex}_{ref} \cdot \left( \frac{\text{HCIC}_{new}}{\text{HCIC}_{ref}} \right)^{-b} \qquad (4)$$

$$\text{Progress Ratio} = (2)^{-b} \qquad (5)$$

$$\text{Learning Rate} = 1 - \text{Progress Ratio} \qquad (6)$$

We consider an S-curve pattern and two scenarios for the deployment of e-SCP plants up to 2070, which ultimately defines the cumulative installed capacity at each year. The SCP core plant is highly modular, which is expected to have a 10–20% learning rate similar to comparable technologies such as electrolysers, batteries, and seawater reverse osmosis desalination plants[63–65]. In the reference scenario, e-SCP plants reach a nominal operational capacity of 48

and 150 $Mt_{SCP}$ $a^{-1}$ in 2050 and 2070, respectively, while the plant's capital cost would decline based on a 10% learning rate. In the advanced scenario, the e-SCP plant installation at each timestep is doubled, and a learning rate of 15% is considered for the SCP core plant capital cost. The nominal capacity of SCP core plant is based on baseload plants with 8000 Full Load hours, while cost-optimised e-SCP plants are expected to have a lower load factor than baseload plants. Thus, the delivered e-protein is expected to be less than the above-mentioned target capacities. Considering a global average achieved Full Load hours of 7300 hours and the 61.75% protein content of SCP, 27.0 and 84.5 $Mt_{protein}$ of e-protein could be produced in the reference scenario in 2050 and 2070, respectively. The global protein demand as food and feed in 2050 is projected to reach about 300 and 1700 Mt, respectively[22,51]. A 27 Mt $a^{-1}$ e-protein supply by 2050 is about 9.0% and 1.6% of projected food and feed protein demand, respectively. These supply rates are well below the limits for e-protein consumption, especially by animals, as 10–19% of animal feed globally (equivalent to 175–307 Mt of microbial protein by 2050) could be substituted by microbial proteins with respect to animals' wellbeing[22].

The detailed fixed operational expenditures (opex) of SCP core plants, including maintenance, insurance, personnel, and outsourced services (cleaning, legal consultation, waste handling, accounting, laboratory analyses, as well as education and training) for the first small-scale and full-scale SCP core plants are provided in the Supplementary Table 2. The energy and mass balance of the e-SCP core plant, as well as its outsourced variable operational costs (such as minerals, process water, wastewater, etc.) are provided in Supplementary Table 3.

For mass deployment of SCP plants beyond 2030, the fixed personnel and outdoor services are projected to decline with a learning rate of 5% and 7.5% in the Reference and Advanced scenarios, respectively. We illustrate the impact of these two scenarios on the capacity and capex of periodic and cumulative installations of e-SCP core plants until 2070 in Fig. 3. The underlying data are also provided in Supplementary Tables 5 and 6. A summary of the techno-economic input data for the core SCP plant used in the simulations is provided in Table 2.

**Onsite flexible hybrid PV-wind power-to-SCP supply chain**
We model semi-flexible onsite Power-to-SCP supply chains (Fig. 2) to evaluate the cost-optimised configuration of the introduced technologies to meet their hourly energy and mass balance in a $0.45° \times 0.45°$ spatial resolution.

The defined power supply technologies include optimally fixed tilted PV, single-axis tracking PV, and wind power plants. The hourly solar irradiation and wind speed in a $0.45° \times 0.45°$ spatial resolution are taken from NASA databases[66,67] and are partly reprocessed by the German Aerospace Centre[68]. The hourly feed-in time series of fixed-tilted PV is based on Gerlach et al.[69] and Huld et al.[70], and the hourly feed-in time series of single-axis tracking PV are based on Afanasyeva et al.[71]. The hourly feed-in time series of wind turbines are according to Gerlach et al.[69], which considers ENERCON standard 3 MW wind turbines (E-101) with 150-metre hub height. A wake effect of 8% is considered on the overall electricity generation of wind farms. Lithium-ion battery and $H_2$-fuelled open cycle and combined cycle gas turbines are introduced to the model as potential daily and seasonal power balancing technologies. The battery system consists of independently scalable battery interface and battery storage for optimal combination of electrical power and energy storage.

A cluster of alkaline water electrolysers is considered for $H_2$ and $O_2$ production at 30 bar. While a single alkaline electrolyser stack is limited in minimum partial load, a large-scale cluster of electrolysers would have a full range flexibility at system level by running down to one stack at its minimum partial load. An $O_2$ processing unit is added as a complementary unit to regular electrolyser plants for the initial handling of required $O_2$, and the extra $O_2$ is vented to the air. The $O_2$ processing unit is based on a design with atmospheric electrolysers and consists of a set of compressors for $O_2$ compression from 1 to 5 bar, small temporarily storage tanks and required piping and balance of plant. Each two stacks share a set of $O_2$ compressor and storage system. As we consider pressurised electrolysers in our system configuration, additional $O_2$ compression is not required. However, in a conservative approach, the cost of initial $O_2$ compressors is not excluded as its cost share in the $O_2$ processing package is unavailable. The produced $H_2$ could be directly used in SCP fermentation tanks and ammonia synthesis unit or could be balanced via the $H_2$ compressor and storage system prior to utilisation.

Three different $H_2$ storage options, namely underground pipe, man-made salt cavern and lined rock cavern are included in the model. Regions with suitable geological formations for salt and rock cavern storage are illustrated in Supplementary Fig. 15. The maximum hourly charge and discharge of all $H_2$ storage options are limited (Supplementary Table 21) to control their pressure change rates and consequently their temperature, as well as tension on the salt cavern storage walls. Consequently, the maximum total hourly $H_2$ storage charging rate throughout the year defines the $H_2$ compressor flow rate. A potentially higher hourly discharge flow rate does not affect the compressor capacity as the discharge is based on internal pressure. The compressor's compression range is based on the maximum potential storage pressure of 150 bar for salt and rock cavern storage. However, most of the time, the storage would not be full and thus would be at lower pressure, which would require less compression work for charging the $H_2$ storage.

For $O_2$ and $CO_2$ storage, gaseous and liquid storage systems are considered in parallel. There are liquefaction units prior to liquid storage tanks. The considered liquefaction units take in gases at 5 bar. The regasification unit at discharge side of the liquid $O_2$ and $CO_2$ is not modelled in the optimisation environment as its cost is considered to be negligible due to surplus of excess heat in the system, which can be used for heating of liquefied gases.

The semi-flexible ammonia synthesis unit is coupled with a cryogenic air separation unit for the required nitrogen[72]. An ammonia storage tank is applied to balance the timing of ammonia production and consumption.

The chiller unit is accompanied by chilled water storage for higher flexibility of power supply to the chiller. In the compound design of chilled water storage, chilled water could be stored at the bottom of the storage tank while the returning cold temperature water from fermentation tanks could be optionally stored at the top of the same storage tank prior to cooling process in the chiller. This way, separate tanks for chilled and cold temperature water are avoided.

High-pressure steam at 10 bar and 180 °C is required in the drum drying units, and at 4.2 bar for pasteurisation and sterilisation. High-pressure steams in the drying unit evaporate most of the water content of the concentrated broth, which leaves the system as water vapour. To simplify the model, all the steam generation is at the higher pressure, while the lower pressure demand could be supplied by a pressure breaker. Two steam generation technologies are introduced to the system. The first option is an industrial electric steam boiler with a maximum output pressure of 12 bar. This option is only suitable for direct generation and consumption of steam. The second option is a steam generation system[73] that decouples steam generation time from power intake time by use of heat storage. In this approach, electric rods are used to heat a heat storage medium in which heat is stored in molten material at ~300 °C. The stored heat could simultaneously or later be used in a steam generation unit with a maximum pressure of 16 bar. The modular charge, storage, and discharge (steam generation) units are defined as separate components in the model to achieve the cost-optimal configuration at each location.

Electric water boilers and heat pumps are the two considered technologies for the supply of heat at 100 °C to the DAC units. The generated heat could be optionally stored in high-temperature heat storage (water storage at 100 °C) prior to consumption by DAC units, which decouples the timing of heat generation and its electricity consumption. The water vapour from the dryer unit is the only considered source of heat for the heat pump. The water vapour is guided to a warm water storage, which acts as the low-temperature heat storage and the heat source for the heat pump. The size of the low-temperature heat storage is based on the mass of liquid water required for heat absorption and condensation of the utilised water vapour. Unutilised heat from the dryer is dumped in a condenser. The cooling systems of waste heat from the dryer are not separately modelled, but their cost is included in the core SCP unit. The considered setup is relatively conservative and further improvements are likely. The water vapour from the drying unit could potentially be directly used by the DAC unit, especially if the DAC ideal regeneration temperature could be reduced by 10 °C to about 90 °C. In addition, the alkaline water electrolysers are expected to develop into high-pressure water electrolysers in the near future, which could also deliver waste heat at higher temperatures that could be directly used by DAC units. All these potential developments could make heat supply to DAC units in Power-to-SCP plants simpler and cheaper.

The hourly flows of minerals and freshwater used by water electrolyser, ammonia solution, and fermentation are not modelled for the sake of simplicity, and their costs are considered on an annual basis in the operational cost.

The semi-flexible e-SCP core plant has a minimum operational load of 50% as well as hourly ramp-up and ramp-down rates of 1% and 0.3%, respectively. In practice, there would be an 80-hour turnover time between feeding feedstocks to the fermentation and production of the SCP solution, as well as delivery of dry SCP after the cell separation and drying stage. However, for the sake of simplicity, the whole processes in the core SCP plant are considered to occur simultaneously on an hourly level. This approach means a certain rate of change in the input feedstocks supply to the SCP core plant would result in the same rate of steam demand for SCP processing and SCP production. Nevertheless, the impact of such simplification is considered marginal as the core SCP plant has a minimum operational capacity of 50% at all times, and the plant tends to run at high utilisation rates in the cost-optimised operation. To provide a baseload supply of SCP, an SCP storage unit is added to balance out the variable production. The SCP storage unit has a prefixed maximum charge and discharging rate, which provides a minimum full charge or discharge time of 7 weeks or 168 hours.

## Hourly energy and mass balance

The model and its constraints are defined in the Matlab software, version R2020a[74]. The base model has been previously used in peer-reviewed scientific publications such as on Power-to-Ammonia[72] and has been expanded for this study. The world is divided into 0.45° × 0.45° nodes, resulting in a 400 × 800 matrix, of which 74,487 nodes represent the land area, and the rest are bodies of water and Antarctica. The techno-economic potential of Power-to-SCP in each node has been evaluated independently based on onsite hourly solar and wind potentials, and the availability of suitable geological formations for salt or rock cavern $H_2$ storage.

The applied hourly energy and mass balance is shown in Eq. (7), which considers hourly generation, consumption, storage, storage loss, and an excess amount of each flow by relevant technologies, as well as the target demand. The only target demand in the optimisation package is an hourly SCP supply (not production) of 1000 kg. Abbreviations: flow type ($f$), technology ($i$), electricity (el), low-temperature heat at 20–80 °C (LTH), high-temperature heat (HTH) at 80–100 °C, very high-temperature heat (VHTH) at 300 °C, chilled water (ChW) at 4 °C, charge (char), and

discharge (disc). All hourly values are equal to or greater than zero.

$\forall h \in [1, 8760] | \forall f \in [\text{el, H2, O2, CO2, NH3, LTH, HTH, VHTH, steam, ChW, SCP}]$ :

$$
\begin{aligned}
& \sum_i^{\text{tech}} \text{Gen}_{f,h,i} - \sum_i^{\text{tech}} \text{Cons}_{f,h,i} - \sum_i^{\text{tech}} \text{storage char}_{f,h,i} \\
& + \sum_t^{\text{tech}} \text{storage disc}_{f,h,i} + \sum_t^{\text{tech}} \text{storage loss}_{f,h,i} \\
& - \sum_t^{\text{tech}} \text{excess}_{f,h,i} = \sum_t^{\text{tech}} \text{target demand}_{f,h,i}
\end{aligned} \tag{7}
$$

Storage loss includes losses at charge and discharge hours as well as hourly self-discharge that are formulated as shown in Eqs. (8, 9). The hourly charge and discharge rates of storage options are constrained, as defined in Eqs. (10, 11). For gas storage options, an additional limiting factor for the charge rate is the maximum flow rate of the compressor before storage. Abbreviations: state of charge (SoC), efficiency (eff), storage for flow type ($fs$), and installed capacity (instCap).

$\forall h \in [1, 8760] | \forall fs \in [\text{el, H2, CO2, NH3, LTH, HTH, VHTH, ChW and SCP storage}]$ :

$$
\begin{aligned}
\text{SoC}_{fs,h} = \text{SoC}_{fs,h-1} \cdot \left(1 - \text{self disc eff}_{fs}\right) - \frac{\text{disc}_{fs,h}}{\text{disc eff}_{fs}} \\
+ \text{char}_{fs.h} \cdot \text{char eff}_{fs}
\end{aligned} \tag{8}
$$

$$
\begin{aligned}
\text{SoC}_{fs,1} = \text{SoC}_{fs,8760} \cdot \left(1 - \text{self disc eff}_{fs}\right) - \frac{\text{disc}_{fs,1}}{\text{disc eff}_{fs}} \\
+ \text{char}_{fs,1} \cdot \text{char eff}_{fs}
\end{aligned} \tag{9}
$$

$$
\text{storage char}_{f,h,i} <= \text{instCap}_i \cdot \text{maximum hourly charge}_{f,h,i} \tag{10}
$$

$$
\text{storage disc}_{f,h,i} <= \text{instCap}_i \cdot \text{maximum hourly discharge}_{f,h,i} \tag{11}
$$

The relevant technologies to each flow type are as follows. Technologies with electricity flow (Tech $f$(el)) include fixed tilted PV power plant, single-axis tracking PV power plant, wind power plant, $H_2$-fuelled open cycle gas turbine, $H_2$-fuelled combined cycle gas turbine, battery interface, battery storage, water electrolyser, $O_2$ processing unit, SCP core plant, ammonia plant, $H_2$ compressor, $O_2$ compressor, $O_2$ liquefaction unit, DAC, $CO_2$ compressor, $CO_2$ liquefaction unit, heat pump, electric water boiler, electric rod, electric steam boiler, and chiller. Technologies with $H_2$ flow (Tech $f$($H_2$)) include water electrolyser, SCP core plant, ammonia synthesis plant, $H_2$ compressor, $H_2$ salt cavern, $H_2$ rock cavern, $H_2$ underground pipe, $H_2$-fuelled open cycle gas turbine, and $H_2$-fuelled combined cycle gas turbine. Technologies with $O_2$ flow (Tech $f$($O_2$)) include water electrolyser, $O_2$ processing unit, SCP core plant (fermenter), $O_2$ compressor, $O_2$ liquefaction, $O_2$(g) storage, and $O_2$(l) storage. Technologies with $CO_2$ flow (Tech $f$($CO_2$)) include DAC, SCP core plant (fermenter), $CO_2$ compressor, $CO_2$ liquefaction unit, $CO_2$(g) storage, and $CO_2$(l) storage. Technologies with $NH_3$ flow (Tech $f$($NH_3$)) include e-ammonia plant, SCP core plant (fermenter), and ammonia storage. Technologies with LTH flow (Tech $f$(LTH)) include {SCP core plant (dryer), LTH storage, and heat pump. Technologies with HTH flow (Tech $f$(HTH)) include DAC, heat pump, electric water boiler, and HTH storage. Technologies with VHTH flow (Tech $f$(VHTH)) include electric rod, VHT heat storage, and decoupled steam generator. Technologies with steam flow (Tech $f$(steam)) include electric steam boiler, decoupled steam generator, and SCP core plant. Technologies with chilled water flow (Tech $f$(ChW)) include chiller, SCP core plant

(fermenter), and chilled-cold water storage. Technologies with SCP flow (Tech $f$(SCP)) include SCP core plant and SCP storage.

## Economic evaluation and cost optimisation

The Eq. (12–14) are used to calculate the levelised cost of SCP according to the guideline by NREL[75]. Abbreviations: capital recovery factor (crf), also known as annuity factor, weighted average cost of capital (WACC), lifetime (N), applied technology (i), capital expenditures per unit of capacity (Capex), annual operational expenditures per unit of capacity (Opex), generation (Gen). A global uniform real WACC of 7% is used in all the calculations, excluding the inflation rate considered in nominal WACC. Assuming an equity share of 30% and an interest rate of 4%, a WACC of 7% would lead to a return on equity of 14%.

$$crf = \frac{WACC \cdot (1 + WACC)^N}{(1 + WACC)^N - 1} \quad (12)$$

Levelised cost of SCP

$$= \frac{\sum_i^{tech}((Capex_i \cdot crf_i + Opex_{fix,i}).instCap_i + Opex_{var,i} \cdot Gen_i)}{Annual\ SCP\ production} \quad (13)$$

$$Gen_i = \sum_{h=1}^{8760} Gen_{i,h} \quad (14)$$

The installed capacity of each technology (i) is set to be equal to or higher than its maximum hourly flow divided by its annual availability rate to account for maintenance time on an annual basis, as formulated in Eqs. (15–16). The minimum operational load and maximum ramping rates for applicable technologies are regulated based on Eqs. (17–19).

$$instCap_i >= \frac{maximum\ hourly\ flow_i}{availability_i} \quad (15)$$

$$instCap_i >= \frac{maximum\ hourly\ SoC_i}{availability_i} \quad (16)$$

$$minimum\ operational\ load: \\ Gen_{i,h} >= instCap_i \cdot availability_i \cdot minimum\ load_i \quad (17)$$

$$ramp\text{-}up\ limit: \\ (Gen_{i,h} - Gen_{i,h-1}) <= instCap_i \cdot rampup\ rate_i \cdot availability_i \quad (18)$$

$$ramp\text{-}down\ limit: (Gen_{i,h-1} - Gen_{i,h}) \\ <= instCap_i \cdot rampdown\ rate_i \cdot availability_i \quad (19)$$

The objective of the model is to deliver a relatively small, predefined baseload hourly SCP supply (Gen$_h$ = 1000 kg h$^{-1}$ · different from flexible SCP production) and meet the hourly energy and mass balance of all components within the system constraints throughout the year at a minimum cost. Consequently, a fixed annual supply of SCP (Gen = 8760 t$_{SCP}$ a$^{-1}$) is met through the flexible operation. To achieve this goal, a linear optimiser (Mosek[76]) is applied to optimise the installed capacities of available technologies and their hourly flows, as shown in Eq. (20).

$$min\left(\sum_i^{tech}((Capex_i \cdot crf_i + Opex_{fix,i}) \cdot instCap_i + Opex_{var,i} \cdot Gen_i)\right) \quad (20)$$

## Theoretical production potential

The average installation density (instDens) of fixed tilted PV and single-axis tracking PV is considered to respectively increase from 91 and 62 MW km$^{-2}$ in 2020 to 137 and 94 MW km$^{-2}$ in 2050 by extrapolation of trends in the past decade[49] and projected PV module efficiency[77]. A fixed average installation density of 8.4 MW km$^{-2}$ is considered for wind power plants from 2020 to 2050 according to Bogdanov and Breyer[78]. A maximum 10% of land at each node could be used for the installation of PV and wind power plants to account for uncertainties with area availability. The chosen optimisation sample (Gen) is relatively small to ensure that PV and wind installed capacities would be within their area limit. Once the optimised installed capacities of PV and wind technologies are identified by the optimiser, all system elements are expanded by a minimum multiplier to the level where PV or wind installed capacity reaches its nodal area coverage limit, as formulated in Eqs. (21–25). The potential nodal cost-optimised e-SCP production is calculated by multiplying the sample annual production by the nodal minimum multiplier, and the global e-SCP production potential is the cumulative nodal e-SCP potential, as shown in Eqs. (26–27). As nodal area differs by latitude, regional production potential per a unified area limit is introduced via Eq. (28). Abbreviations: fixed-tilted PV (fixed tilt), single-axis tracking PV (1-axis), circumference (circum), node (n).

$PV$ instDens

$$= \frac{instDens_{fixed\ tilt} \cdot instCap_{fixed\ tilt} + instDens_{1-axis} \cdot instCap_{1-axis}}{instCap_{fixed\ tilt} + instCap_{1-axis}} \quad (21)$$

$$Nodal\ area = \left(\frac{polar\ circum}{2} \cdot \frac{0.45°}{180°}\right) \cdot \\ \left(equatorial\ circum \cdot \frac{0.45°}{360°} \cdot Cosine(latitude°)\right) \quad (22)$$

$$PV\ multiplication\ factor = \frac{PV\ instDens \cdot nodal\ area \cdot 10\%\ area\ limit}{instCap_{fixed\ tilt} + instCap_{1-axis\ PV}} \quad (23)$$

$$wind\ multiplication\ factor = \frac{wind\ instDens \cdot nodal\ area \cdot 10\%\ area\ limit}{instCap_{wind}} \quad (24)$$

$$minimum\ multiplier \\ = Min(PV\ multiplication\ factor, wind\ multiplication\ factor) \quad (25)$$

$$Nodal\ production\ potential = minimum\ multiplier_n \cdot Gen_n \quad (26)$$

$$Global\ SCP\ production\ potential = \sum_n^{node} minimum\ multiplier_n \cdot Gen_n \quad (27)$$

$$Regional\ production\ potential\ per\ km^2 = \frac{Nodal\ production\ potential}{nodal\ area} \quad (28)$$

For water electrolysers, a stack replacement cost is included only if it reaches its operational lifetime before the system-level calendar lifetime of 30 years. The stack replacement cost is considered as a conditional variable cost based on stack cost in 15 years after initial plant installation and its utilisation rate until the 30-year system lifetime is reached.

## Data availability

Additionally, the data supporting the findings of this study are available in the Supplementary Information document. Source data are provided with this paper.

## Code availability

The main model code is available from the authors upon request.

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

## Acknowledgements

The authors gratefully acknowledge the public funding of Business Finland for the 'P2XENABLE' project under the number 8588/31/2019 and the LUT University Research Platform 'GreenRenew', which partly funded this research. We also thank Gabriel Lopez for proofreading.

## Author contributions

M.F. and C.B. designed the study. M.F. developed the model, generated the results, and wrote the paper. F.J. and M.F. carried out research on conventional protein demand, price, and environmental impacts. P.V. and P.T. provided data for the Solar Foods plant. C.B. supervised the work and reviewed the research and manuscript.

## Competing interests

P.V. is the CEO and co-founder of Solar Foods. Petri Tervasmäki is employed by Solar Foods. The remainig authors declare no competing interests.
