## [Peer Review File · Nature Communications]

REVIEWER COMMENTS

Reviewer #1 (Remarks to the Author):

Overview

This manuscript describes an economic analysis to evaluate the global potential of Single Cell Protein (SCP) production based on a production process by the start-up company SolarFood. In the introduction, the authors discuss the environmental challenges associated with increasing protein consumption and their production based on conventional agriculture. They propose SCP produced from sustainable energy sources and feedstock as an environmentally friendly alternative. The process uses chemoautotrophic H₂-oxidising bacteria. Hydrogen (H₂) and oxygen (O₂) are provided by water electrolysis, CO₂ is captured by DAC units, and ammonium is supplied by a Power-to-Ammonia plant. Accordingly, electricity is the main provider of feedstock. The study evaluates the global potential of SCP production using renewable energy sources, such as hybrid photovoltaics-wind power plants. The manuscript outlines a potential roadmap for industrial-scale SCP production, starting with 128kt/a in 2028 and aiming to reach production capacities of 37 Mt/a by 2050 and 93 Mt/a by 2070. In chapter 1 of the results, the authors determined the total capex of a SCP core plant, which will decline until 2070 due to economy of scale. In chapter 2, the authors analyzed the availability of renewable energy. PV-dominated regions will generate electricity at the lowest price by 2050, emerging as the best sites for cheap e-protein production. Because of the declining capex of the SCP and economy of scale, the cost of renewable electricity-based protein (e-protein) could significantly decrease globally to 2.1-2.3 €/kg in 2035 and 1.6-1.9 €/kg in 2050 (chapter 3). In chapter 4, the authors discuss the need for key components and balancing technology and their electricity consumption in detail for seven selected sites. Finally, they analyzed the area demand of hybrid PV-wind Power-to-SCP systems with the result that only ~0.017 % of the global land area will be needed to cover 28 % of the worldwide protein demand as food with e-protein in 2070. The manuscript highlights that e-proteins can help mitigate global warming potential and other environmental impacts associated with traditional food production methods. Notably, the study shows that e-protein production has the potential to become competitive with protein production via conventional agriculture if prices for electricity generation and balancing technologies decline globally.

Language

Overall, the paper presents valuable information; however, it could benefit from significant improvements to enhance readability and clarity.

1) Sentence Length: Some sentences are overly long, making it challenging for the reader to follow the text. Consider breaking them down into smaller, more digestible sentences to improve readability.

- 2) **Argument Clarity:** The argumentation in certain sections (especially section: Power-to-SCP system analysis at selected sites) is not as straightforward as it could be. Providing concise and logically structured arguments is essential to guide the reader through the paper more effectively. Please add a summary in the discussion naming the study's main findings. It helps to understand key improvements needed to make the Power-to-SCP technology competitive with soy and pea isolates.
- 3) **Figures:** Please include references to figures and diagrams. It helps the reader to connect the visual information with the corresponding explanations and makes the information more accessible.
- 4) **Proofreading:** Ensure that the paper is thoroughly proofread to correct any typographical errors that might hinder comprehension: f.e. line 118: draught; the reviewer, however, thinks “droughts” is meant in this context.
- 5) The TEA is as good as its assumptions. To this reviewer, some of the numbers are looking really good. The eventual price of renewable electric energy is cheap in this study. The price of the produced protein is very low and even the amount of electric energy that would be required to produce one kg of protein is relatively low. The study could benefit from a Table where assumptions and results are compared to other TEA studies of protein production (at least 2-3 TEAs were mentioned as references). Here, power-to-protein systems can be compared, but also other SCP systems such as Quorn, or other proteins, including those based on soy. It would be good to compare numbers from other studies and then explain why the numbers are lower/higher etc. in your study. I am proposing a Table of final and important values to compare to other studies. Table L is too complicated for such a comparison.
- 6) DAC is a very energy-intensive process due to the low concentrations of CO₂ and the thermodynamic penalty of taking CO₂ off for example an amine. In this study, the electric energy cost of removing CO₂ from the air seems relatively low. How valid are your assumptions here, especially since no real scaled-up DAC system exists? Again, please compare your TEA to several TEAs on the matter, including the TEAs that are critical to DAC. I noted Table L, but for the production plant and DAC no other references were given as a comparison. How good are your data and assumptions here, this is difficult to gauge for a reviewer.
- 7) For calculating the energetic cost of the Solar Foods production (hydrogen formation via electrolysis and hydrogen consumption by the culture), is there a thermodynamic calculation present based on first principles, or is the data taken from a pilot plant? How good is it? Again, please compare it to other studies or technologies.
- 8) What is exciting to share is to amount of land that is required, which is very low. The cost of the protein is also very low, but how can we ensure you do not overpromise? Ultimately, we need several TEAs to show the promise, but we do not have them for this exact technology. It is still helpful to compare it to several “next close” technologies and discuss the differences.

Figures

1) The reviewer suggests numbering the figures with A, B, C,... instead of (left), (right) because it simplifies referring to the figures in the text

2) Please provide larger axis labels and descriptions in the figures

Comments

- L15: please add “per year”

- L17: In the chapter “Levelised cost of e-protein” numbers for 2050 are 1.6 – 1.9 €/kg(protein), why is the number different in the abstract?

- L30: Reference 7 Kim et al. 2019 name that Animal-source foods contribute 18% of calorie

Please remove comma before globally.

- L35: only use “to” instead of “in order to”, remove comma before to: ...to a sustainable one to be in line with the United Nations...

- L38: please add comma after “in particular,”; remove “s” protein supply.

Please start a new sentence for “which their availability...”

- L40: Please change sentences: One of the most effective conventional approaches to mitigate the food system’s GHG emissions is switching to a plant-based diet.

- L41: Please exchange “diet” with “switch”

- L42: remove comma before “nor”

- L47: please delete: “by the use of microorganisms” as this is already implied by microbial protein

- L51: Why are only B-vitamins present? The reviewer suggests generalizing and using the term micronutrients, which is referred to as vitamins and minerals in general.

- L52: Protein has no doubling time but bacterial cells. Please correct the sentence

- L52: The reviewer thinks growth method is the wrong wording; better use “energy conversion pathways” or similar term

- L58: Please correct to sulfide

- L61: The reviewer assumes that the authors want to say: Single Cell Protein is not a fully new concept.

- L61: Please correct to: Germany produced yeast biomass on large scale...

- L63: Please add production in this sentence as microbial biomass is not a food system, change to: CO₂-based microbial biomass production was considered as a circular food system...

- L72-L75: Divide sentence in two: Moreover, direct air capture (DAC) of CO₂ is emerging as an affordable technology for sustainable CO₂.

- L79-L89: The reviewer feels this paragraph is just a sequential list of sources. The authors are kindly requested to categorize this section into specific research domains and guide readers on how these listed research sources are connected to the content presented in this paper.
- L90: What do the authors explicitly mean by “by use of electricity”, baseload already implies the minimum amount of electric power needed.
- L94: Please explain the abbreviation RE (renewable energy)
- L96: Please change “baseload of”
- L97: Can the authors please indicate the volume of the 100 fermentation tanks
- L105: Can the authors please indicate if the remaining 5 % is water or medium.
- L112: Please add a short description and explanation of abbreviations in the table header.
- L118: Please add a comma before “,such as”. The reviewer believes “droughts” instead of “draughts” is meant in this context
- L119: The reviewer believes that the authors mean: maximises the efficiency in using nutrients
- L121: the reviewer wonders what “energy system space” means, can the authors please define this term
- L122: The authors believe that “substrate shortage” is a better wording
- L123: Please explain the problem of current fluctuations of the different renewable energies here, and what balancing technologies are needed.
- L130: Please always introduce abbreviations: Photovoltaics (PV)
- L144-L147: Please simplify this sentence as, in the opinion of the reviewer, the rationale for the study becomes not clear
- L156: “r” is missing, please correct to “learning rate”
- L164: The reviewer thinks that figure 3 (left) is not needed in the main text, because it is not mentioned in the text. The reviewer suggests moving the figure to the supplementary information.
- The reviewer does not understand the axis scale of the blue bars (installed capacity Ref. scenario; additional installed capacity Adv. scenario) as the numbers do not match the right y-axis scale. The change from 2028, 2030 and 2035 is barely readable. The reviewer asks to rescale the axis and enlarge bars.
- L169: correct small-scale
- L172: The term “learning by doing” seems wrong here, please change to “learning curve effect”
- L178: The reviewer does not understand the term “regional potential” as in this context it seems to depend on “regional availability of renewable energy”
- L180: please introduce the abbreviation: photovoltaic (PV)

- L193: Please pay attention to the same terminology in text, subtitle, and figure as it facilitates better comprehension for the reader: cost of electricity supply to e-SCP
- L198: Please add (e-protein) after “cost of its protein content” to have a connection to the figure and for easier understanding
- L204: The reviewer thinks it is better to use the term “Capex” instead of “cost of the SCP core plant”, as the term Capex was also used in the previous chapter
- L207: The reviewer asks if the supplementary figures can be labeled uniformly. A different numbering as in the main text is recommended for supplementary figures.
- L213: Please restructure the whole chapter as the argumentation is not clear to the reviewer, please also refer to Fig. 6 in the text to connect the explanations with the visual information and use the same wording in the text as in the caption of the figure (the reviewer was wondering what are parts of the Power-to steam system L220 in figure 6); explain abbreviations when mentioned for the first time
- L238: The reviewer does not understand the sentence. Can the authors please explain why higher installed capacities are needed in PV-dominated regions compared to wind-dominated areas
- L260: the authors were comparing different costs in different regions before; this information is, however, missing for the non-energetic costs. The reviewer was wondering if these costs are also similar in all areas.
- L273: The reviewer thinks Fig. 6 (bottom; left, middle, right) are not mentioned and explained in the main text; please indicate in the text if data is explained by referring to the figure.
- L282-L286: Please simplify this sentence and divide the sentence into several short sentences
- L286 and the following: Please exchange “/” by “or” because the authors also use “/” division symbol, making the sentence hard to understand.
- L292-306: The reviewer suggests moving this paragraph to the previous chapter as it does not fit in the chapter “area demand”
- L298: The reviewer is wondering where to find the data of over 80 MWh/t(protein) in Far East Russia, please include a reference to the data here or include (data not shown)
- L309: The reviewer wonders what is meant with “here”; please add a reference to the data
- L378: Please add label next to the y-axis
- L387: Please add “per year; /a” in the text
- L398: Please change: aiming SCP supply at industrial scale
- L414: correct “progress ratio”

References

- Please correct references towards consistency; Journal pages are often missing.

Reviewer #2 (Remarks to the Author):

In the research paper on single-cell protein (SCP) plants, several remarkable strengths shine through, underlining the significance and rigor of the study. First and foremost, the paper distinguishes itself by offering an incredibly detailed and comprehensive examination of SCP technology. It delves deep into the intricate technical facets, leaving no stone unturned. This meticulous approach ensures that readers gain a profound understanding of SCP plants, making it a valuable resource for researchers and industry professionals alike. One of the paper's standout strengths is its inclusion of a rigorous techno-economic assessment. This adds a layer of credibility and practicality to the research, allowing stakeholders to evaluate the feasibility and economic viability of SCP plant implementation. Such assessments are crucial in bridging the gap between theoretical concepts and real-world applications. Furthermore, the paper's forward-looking perspective is commendable. By projecting SCP plant deployment trends up to the year 2070, it provides insights into the long-term evolution of this technology. This foresight is invaluable, especially in the context of sustainable food production and environmental considerations. The paper distinguishes itself with its complex modeling approach, involving intricate calculations for hourly energy and mass balances within the hybrid PV-wind power-to-SCP supply chain. This level of scientific rigor serves as a robust foundation for evaluating the proposed system's feasibility. Moreover, the paper's scope is vast, covering an array of technologies, from solar and wind power generation to hydrogen production and ammonia synthesis. This allows grasping the intricate interplay of technologies in the SCP plant ecosystem, emphasizing the complexity of the subject matter.

However, to enhance accessibility, the paper could benefit from simplifying explanations for readers with varying levels of expertise.

Reviewer #1 (Remarks to the Author):

Overview

This manuscript describes an economic analysis to evaluate the global potential of Single Cell Protein (SCP) production based on a production process by the start-up company SolarFood. In the introduction, the authors discuss the environmental challenges associated with increasing protein consumption and their production based on conventional agriculture. They propose SCP produced from sustainable energy sources and feedstock as an environmentally friendly alternative. The process uses chemoautotrophic H₂-oxidising bacteria. Hydrogen (H₂) and oxygen (O₂) are provided by water electrolysis, CO₂ is captured by DAC units, and ammonium is supplied by a Power-to-Ammonia plant. Accordingly, electricity is the main provider of feedstock. The study evaluates the global potential of SCP production using renewable energy sources, such as hybrid photovoltaics-wind power plants. The manuscript outlines a potential roadmap for industrial-scale SCP production, starting with 128kt/a in 2028 and aiming to reach production capacities of 37 Mt/a by 2050 and 93 Mt/a by 2070. In chapter 1 of the results, the authors determined the total capex of a SCP core plant, which will decline until 2070 due to economy of scale. In chapter 2, the authors analyzed the availability of renewable energy. PV-dominated regions will generate electricity at the lowest price by 2050, emerging as the best sites for cheap e-protein production. Because of the declining capex of the SCP and economy of scale, the cost of renewable electricity-based protein (e-protein) could significantly decrease globally to 2.1-2.3 €/kg in 2035 and 1.6-1.9 €/kg in 2050 (chapter 3). In chapter 4, the authors discuss the need for key components and balancing technology and their electricity consumption in detail for seven selected sites. Finally, they analyzed the area demand of hybrid PV-wind Power-to-SCP systems with the result that only ~0.017 % of the global land area will be needed to cover 28 % of the worldwide protein demand as food with e-protein in 2070. The manuscript highlights that e-proteins can help mitigate global warming potential and other environmental impacts associated with traditional food production methods. Notably, the study shows that e-protein production has the potential to become competitive with protein production via conventional agriculture if prices for electricity generation and balancing technologies decline globally.

We would like to specifically thank the reviewer #1 for the detailed and constructive comments.

We would like to offer some clarifications regarding the Overview comment:

- “The manuscript outlines a potential roadmap for industrial-scale SCP production, starting with 128kt/a in 2028 and aiming to reach production capacities of 37 Mt/a by 2050 and 93 Mt/a by 2070.”

In the initial submission, the operational production capacity in 2028 was 12.8 kt_{SCP}/a (7.9 kt_{protein}/a) that reached to 60 Mt_{SCP}/a (37 Mt_{protein}/a) in 2050 and 150 Mt_{SCP}/a (93 Mt_{protein}/a) in 2070. Such data was available in Supplementary Tables F and G. Solar Foods has already achieved a productivity of 1.0 g_{CDW}/L/h compared to 0.8 g_{CDW}/L/h that was used in the initial submission. Thus, for the same volume of the fermentation tanks, we have increased the production of the first plant in 2028 to 16.4 kt_{SCP}/a (10.1 kt_{protein}/a). To have a more conservative Base Scenario, we opted for a lower compound annual growth rate in early years which lowered the production capacity to 48 Mt_{SCP}/a (30 Mt_{protein}/a) in 2050. The target capacity in 2070 remains at 150 Mt_{SCP}/a (93 Mt_{protein}/a).

- “Finally, they analyzed the area demand of hybrid PV-wind Power-to-SCP systems with the result that only ~0.017 % of the global land area will be needed to cover 28 % of the worldwide protein demand as food with e-protein in 2070.”
Sorry if our framing of this part was confusing. Since we did not have a projection of protein demand as food and feed supply in 2070, we stated the ratio of e-protein production in 2070 to protein demand as food in 2050 to be 28%. To better show the scale of area demand, we have now updated this part as follows: “... theoretically, supplying 100% of the global protein demand as food and feed in 2050 by e-protein would require 0.54% of the global land area for respective solar power generation, which could even be on uncultivable land.”
- We have adopted some more conservative techno-economic input data, which in return have increased the stated e-protein production cost at each timestep. The whole revised R1 manuscript is updated accordingly.

Language

Overall, the paper presents valuable information; however, it could benefit from significant improvements to enhance readability and clarity.

1) Sentence Length: Some sentences are overly long, making it challenging for the reader to follow the text. Consider breaking them down into smaller, more digestible sentences to improve readability.

Indeed. Long sentences have been mostly simplified by breaking them down to smaller sentences wherever possible.

2) Argument Clarity: The argumentation in certain sections (especially section: Power-to-SCP system analysis at selected sites) is not as straightforward as it could be. Providing concise and logically structured arguments is essential to guide the reader through the paper more effectively.

The objectives of the mentioned section, now titled “Cost distribution and system dynamics at selected sites” are: firstly, to identify the main contributors to the energy consumption and final cost of e-protein; secondly, to provide an insight on how the sub-units operate and interact in the cost-optimised system, such as the choice of competing technologies and their utilisation rate; thirdly, to put the scale of the required components, particularly power and hydrogen supply units, into perspective. Such insights help to identify the key factors for further improvements and sensitivity analyses. Moreover, such details would expand the bases for a more detailed analysis and debate on the input data and results of this study by readers.

This section has been reworked to better support these goals.

We have used the information in this section in our responses to some of other comments, which emphasises the need for this section to understand the whole system.

Please add a summary in the discussion naming the study's main findings. It helps to understand key improvements needed to make the Power-to-SCP technology competitive with soy and pea isolates.

According to the journal’s guideline for authors, a conclusion or summary of the results is not part of the structure of the article. Even the Discussion section is optional and should be succinct. In other words, the Discussion section should be maximum 2 small paragraphs. For that reason, unfortunately, it is challenging to allocate a separate paragraph to the summary of the main findings. However, we think that our Discussion section is already a summary of the main findings and a reflection on them

in comparison with soy and pea isolates. This should be now more apparent in the revised Discussion section. For example, the e-SCP cost range, its land use, and energy and water demands are provided and compared with other alternatives in Figure 8 and discussed accordingly. e-SCP would be already cost competitive with soy and pea isolates by 2028 in the base scenario. However, additional measures are required for e-SCP competitiveness with soybean as a feed. The improvements required for becoming competitive with soybean as a feed are provided in the paragraph before Figure 8.

3) Figures: Please include references to figures and diagrams. It helps the reader to connect the visual information with the corresponding explanations and makes the information more accessible.

We have now included references to figures and sub-figures in the main text.

4) Proofreading: Ensure that the paper is thoroughly proofread to correct any typographical errors that might hinder comprehension: f.e. line 118: draught; the reviewer, however, thinks “droughts” is meant in this context.

Thank you for pointing this out. The revised manuscript has been proofread by a native English speaker. The specifically mentioned typo has been fixed as well.

5) The TEA is as good as its assumptions. To this reviewer, some of the numbers are looking really good. The eventual price of renewable electric energy is cheap in this study. The price of the produced protein is very low and even the amount of electric energy that would be required to produce one kg of protein is relatively low. The study could benefit from a Table where assumptions and results are compared to other TEA studies of protein production (at least 2-3 TEAs were mentioned as references). Here, power-to-protein systems can be compared, but also other SCP systems such as Quorn, or other proteins, including those based on soy. It would be good to compare numbers from other studies and then explain why the numbers are lower/higher etc. in your study. I am proposing a Table of final and important values to compare to other studies. Table L is too complicated for such a comparison.

5.1. Cost of renewable electricity:

According to the IEA PVPS 2021 report, 60 GW of utility-scale solar PV system were installed globally in 2020 at a price range of 560–600 USD/kW (458–500 €/kW for a long-term €/USD exchange rate of 1.2), averaging at 479 €/kW. This value well matches our 2020 capex assumption of 475 €/kW in 2019€ value for fixed tilted PV systems.

link to the IEA report: https://iea-pvps.org/trends_reports/trends-in-pv-applications-2021/

The long-term capex projection of PV power plants is from peer-reviewed literature based on the solar PV learning curve which does not show an indication of breaking the trendline. In fact, the applied capex projection might be too conservative compared to current developments. For example, just in one year from April 2023 to April 2024 the European spot market price of mainstream solar PV modules (efficiency up to 22%) declined by 55% to 130 €/kW (<https://www.pvxchange.com/Price-Index>). Our assumed PV system capex in 2025 is 370 €/kWp, while the PV module is the costliest part of a large utility-scale PV power plant.

It is also worth mentioning that the required PV capacity for a full-scale Power-to-SCP plant is between 2 to 6 GW in most regions. Only a handful of PV power plants of such scale exist in the world today. Such giga-scale projects are expected to benefit from the economy of scale and have lower capex than

the 'normal' utility-scale PV plants as we know today. In addition, since our capex numbers are in 2019€ value, to compare current prices with our capex projections, the inflation from 2019 to 2023 should be considered.

In terms of the levelised cost of electricity, we report on a range of 15–25 €₂₀₁₉/MWh by PV power plants at best sites in 2030 (Figure 4). However, tenders of large utility-scale PV power plants in Portugal, Chile, Qatar, and Saudi Arabia have been already between 10-20 USD₂₀₂₂/MWh in the recent past, according to the PV Trends report of the PV Systems Programme of the International Energy Agency (https://iea-pvps.org/trends_reports/trends-2023/).

Wind power capex is based on a report for the European Commission. In other parts of the world, in particular in India and China, the cost is typically below the European benchmark.

Link to the report: <https://op.europa.eu/en/publication-detail/-/publication/599a1d8e-509a-11eb-b59f-01aa75ed71a1/language-en>

We have also added a series of sensitivity analyses to the Supplementary Information, including the impact of 10% higher and lower PV and wind capex on the final cost of e-protein in 2030.

5.2. TEA comparison with other e-SCP studies:

A new section *“Techno-economic comparison of Solar Foods e-SCP plant and literature”* has been added to the Supplementary Information. It includes two tables summarising the key specifications and results of this study compared to relevant literature. In addition, the differences are extensively discussed in the section.

5.3. TEA comparison with other types of SCPs:

We wish to keep a techno-economic comparison of e-SCP to the literature on other types of SCPs and plant-based proteins out of the scope of this study. A decent analysis of such kind with explanation of the roots of differences could be a full review paper on its own. Thus, we think it would be more helpful if an independent group of researchers with comparable knowledge of all relevant technologies perform such a TEA in a later study.

6) DAC is a very energy-intensive process due to the low concentrations of CO₂ and the thermodynamic penalty of taking CO₂ off for example an amine. In this study, the electric energy cost of removing CO₂ from the air seems relatively low. How valid are your assumptions here, especially since no real scaled-up DAC system exists? Again, please compare your TEA to several TEAs on the matter, including the TEAs that are critical to DAC. I noted Table L, but for the production plant and DAC no other references were given as a comparison. How good are your data and assumptions here, this is difficult to gauge for a reviewer.

The reviewer has raised a fair point. We have now added a new section titled *“S6. Energy and cost projection of solid sorbent Direct Air Capture”* to the Supplementary Information wherein we explain our assumptions for DAC specifications. The section also includes Supplementary Table L, which lists the specifications of solid sorbent DAC plants in recent influential publications. We have also tried to provide all relevant data on existing and near future DAC plants by Climeworks, the developer of the biggest solid sorbent DAC plants to this day. Accordingly, DAC specifications in our study have been updated, with capex and electrical power demand increasing significantly. Consequently, all the simulations have been redone and new results have been generated.

7) For calculating the energetic cost of the Solar Foods production (hydrogen formation via electrolysis and hydrogen consumption by the culture), is there a thermodynamic calculation present based on first principles, or is the data taken from a pilot plant? How good is it? Again, please compare it to other studies or technologies.

We have now addressed these matters in the newly added section “*Techno-economic comparison of Solar Foods e-SCP plant and literature*”. In short, Solar Foods does not disclose full detail of their design at this point. The plant specifications in this study are a combination of: 1. values already achieved in Solar Foods’ pilot plant (such as feed gases utilisation rate and bioreactor productivity rate); 2. values achieved in other comparable experimental studies (such as O₂/CO₂ consumption ratio, which consequently affects the H₂ consumption rate); 3. expected values for large-scale plants based on quotes from industrial suppliers and calculations of three engineering consultant companies (such as capital cost and energy demand of bioreactors and drum dryer). The cost and stoichiometric and empirical energy and mass balance of Solar Foods and other lab-scale or pilot projects are now presented in Supplementary section S3 and further discussed in the respective section.

For energetic cost of hydrogen formation via electrolysis, the Supplementary section S5 (Cost projection of alkaline water electrolyser) has been expanded to provide more information on the basis for the considered capex for water electrolyser. The current efficiency of water electrolyser is based on the quote from a European manufacturer. In addition, Supplementary Table K is added for comparison of our assumptions with short-term and long-term projections in other studies.

8) What is exciting to share is to amount of land that is required, which is very low. The cost of the protein is also very low, but how can we ensure you do not overpromise? Ultimately, we need several TEAs to show the promise, but we do not have them for this exact technology. It is still helpful to compare it to several “next close” technologies and discuss the differences.

We have tried to provide the underlying techno-economic data for all calculations in the form of Supplementary Information. Additionally, the section “Cost distribution and system dynamics at selected sites” in the main text helps the readers to navigate themselves among the results and how they interconnect. Thus, the origins of any result that might be perceived as unexpected should be traceable.

For example, the land requirement is mainly affected by the energy and mass balance (provided by Solar Foods and explained in the Supplementary section S3), the area density of PV and wind installation (improving over time as has been documented for the past), as well as their regional yield and curtailment based on weather data and Power-to-SCP system configuration. If the land requirement is perceived as very low, one could check if they disagree with any of the underlying data. If not, the results could be considered as new insight, based on the assumptions for the energy and mass balance and individual technologies.

The cost of e-protein production is mainly affected by the cost of renewable electricity, SCP core plant, H₂ supply, and DAC. Three engineering consulting companies performed independent design and energy and cost estimation for a Solar Foods 2028 small-scale core plant. We have now modified our assumptions to the most conservative one among the three cases. Consequently, the capital cost, energy demand, land use, and the cost of final product have increased in the revised R1 manuscript compared to the initial submission. The capex of the small-scale SCP core plant in 2028 is 53 times higher than the only available capex in the literature for a comparable size plant. The capex of the first

full-scale SCP core plant (10 times larger than the small-scale plant) declines 40%. This is comparable to an overall scaling factor of 0.77, which we consider reasonable, also in comparison to similar industries. The capex of the full-scale plant declines by 57% for mass deployment from 2030 to 2050 for a learning rate of 10%. Of course, the projected deployment of SCP plants by 2050 is debatable. We find that an achievable scenario given the deployment rate of modular technologies in the past. We consider a 10% learning rate for such modular technology which could be relatively low.

Justifications for the cost of renewable electricity, electrolysers, and DAC plants are provided in response to earlier comments and in the Supplementary Information. While the capex projection for DAC plants is the most uncertain, it has a small impact on the overall cost of e-protein among the above-mentioned factors.

We include the additional costs associated with use of variable sources of power. For example, as we show in Fig. 6b, the hydrogen balancing cost (hydrogen compressor and storage) could be higher than the non-energetic cost of water electrolyser in some regions, such as on the west coast of Finland. To the best of our knowledge, such details are not accounted for in the existing literature. In addition, the cost distribution in Figure 6b shows that any divergence in cost assumption of other technologies would not have a significant impact on the production cost of e-SCP and the conclusions. Accordingly, we have now added a series of sensitivity analyses regarding the impact of solar PV, wind power, electrolyser, DAC, and SCP core plant capex on the final cost of e-SCP in the Supplementary Information section S8.

While some costs might have gone unnoticed, opportunities have also been identified to lower the cost of the system that are not included in this cost analysis. For example, new plant designs are being considered to directly use the waste heat (water vapour) from dryers in DAC units. This would eliminate the need for heat pumps in the system. In addition, the cooling load of bioreactors could be used as a heat source for high temperature heat pumps to provide the required steam in the system, significantly reducing the power demand for cooling and steam generation. Solar Foods have improved the productivity of the bioreactors from 0.8 g_{CDW}/L/h in 2019 to 1.0 g_{CDW}/L/h in 2023. It is expected that such a trend would continue with further optimisation of the process. However, we do not consider any further improvement of the productivity for the plants in 2028 and 2030. An increased productivity would further reduce the capex and energy demand.

A techno-economic comparison to the existing literature on Power-to-SCP has been added to the Supplementary section S3. We wish not to extend the techno-economic comparison to “next close” technologies as it would be out of the scope of this study and our core expertise.

Figures

1) The reviewer suggests numbering the figures with A, B, C,... instead of (left), (right) because it simplifies referring to the figures in the text

The alphabetical numbering of sub-figures has been applied in the revised manuscript.

2) Please provide larger axis labels and descriptions in the figures

The font size of the axis labels and descriptions in Figures 3-7 have been enlarged. For an accepted manuscript for publication, we will also provide the vector file of figures to the journal for better standardisation of the font size and type at publishing stage.

Comments

- L15: please add “per year”

Added. Thanks for noticing.

- L17: In the chapter “Levelised cost of e-protein” numbers for 2050 are 1.6 – 1.9 €/kg(protein), why is the number different in the abstract?

This is because in the abstract we reported on the production cost at the best sites in the world (1.5–1.7 €/kg,protein). The sentence in the main text reported achievable cost levels in all continents (1.6–1.9 €/kg,protein), which would be at a higher cost range. The values are now updated based on new simulation results. The respective text in the chapter “Levelised cost of e-protein” has been also modified to avoid any confusion on that matter, as follows: “By 2050, low-cost e-protein could be produced for 2100–2300 €/t_{protein} at best sites and may be vastly available in all continents at 2300–2500 €/t_{protein} (Fig. 5d)”

- L30: Reference 7 Kim et al. 2019 name that Animal-source foods contribute 18% of calorie Please remove comma before globally.

The calorie share is adjusted to 18%. Thanks for noticing.

The comma has been removed.

- L35: only use “to” instead of “in order to”, remove comma before to: ...to a sustainable one to be in line with the United Nations...

Adjusted.

- L38: please add comma after “in particular,”; remove “s” protein supply. Please start a new sentence for “which their availability...”

Both done.

- L40: Please change sentences: One of the most effective conventional approaches to mitigate the food system’s GHG emissions is switching to a plant-based diet.

Done.

- L41: Please exchange “diet” with “switch”

Done.

- L42: remove comma before “nor”

Comma removed.

- L47: please delete: “by the use of microorganisms” as this is already implied by microbial protein

Deleted.

- L51: Why are only B-vitamins present? The reviewer suggests generalizing and using the term micronutrients, which is referred to as vitamins and minerals in general.

The suggested term ‘micronutrients’ has been adopted.

- L52: Protein has no doubling time but bacterial cells. Please correct the sentence

True. The sentence is now corrected. Thanks for noticing.

- L52: The reviewer thinks growth method is the wrong wording; better use “energy conversion pathways” or similar term

Changed the wording from ‘growth method’ to ‘energy conversion pathways’.

- L58: Please correct to sulfide

‘sulfide’ is the American English spelling. Throughout the paper, we have tried to follow the British English spelling. Thus, we tend to keep the spelling as ‘sulphide’.

- L61: The reviewer assumes that the authors want to say: Single Cell Protein is not a fully new concept. The wording updated to ‘SCP is not a fully new concept.’

- L61: Please correct to: Germany produced yeast biomass on large scale...

Corrected.

- L63: Please add production in this sentence as microbial biomass is not a food system, change to: CO₂-based microbial biomass production was considered as a circular food system...

True. Corrected.

- L72-L75: Divide sentence in two: Moreover, direct air capture (DAC) of CO₂ is emerging as an affordable technology for sustainable CO₂.

The sentence has been updated as suggested.

- L79-L89: The reviewer feels this paragraph is just a sequential list of sources. The authors are kindly requested to categorize this section into specific research domains and guide readers on how these listed research sources are connected to the content presented in this paper.

Sorry if it is not apparent, but the intention of this paragraph is to show the wide aspects of SCP production studies, and where our study would be positioned at. This wording and the order of mentioned studies have been slightly modified to better reflect the intention behind this paragraph.

The paragraph starts with a statement about renewed interest in SCP. The papers mentioned thereafter narrow down the topic of SCP production step by step:

The first reference (Ritala et al. (2017)) covers a review SCP production from various organisms.

The next two references (Linder (2019) and Nyssölä et al. (2022)) narrow it down to microorganisms.

Then Sillman et al. (2019) narrows SCP production further down to H₂-oxidising bacteria by use of renewable electricity, CO₂ by DAC, and H₂ by water electrolyser. The next reference (Givirovskiy et al. (2019)) goes one step further by using in-situ electrolyser instead of feeding H₂ to the reactor. The next reference (Ruuskanen et al. (2021)) shows the results of a pilot-scale project of such concept. And at the end, the two available studies on lifecycle assessment of Power-to-Protein are mentioned.

- L90: What do the authors explicitly mean by “by use of electricity”, baseload already implies the minimum amount of electric power needed.

“by use of electricity” was meant to describe the primary source of energy for the process to differentiate it from a system running on fossil fuels. To address your concern, we have now changed the wording to “Figure 1 illustrates the schematic of a baseload Power-to-SCP production chain,...”

- L94: Please explain the abbreviation RE (renewable energy)

Added to the figure caption.

- L96: Please change “baseload of”

After consulting the proofreader, we modified the caption of Figure 1 to “Power-to-SCP chain for a proposed full-scale e-SCP plant in 2030 for baseload operation”.

- L97: Can the authors please indicate the volume of the 100 fermentation tanks

The volume of each tank is 200 m³. This is now also added to the revised manuscript.

- L105: Can the authors please indicate if the remaining 5 % is water or medium.

The residual moisture includes the non-volatile compounds of the growth medium that is fed to the drying unit. This includes some residual salts that are not incorporated into the biomass during cell growth. The residual salts are less than 4% of the total moisture, and consequently less than 0.2% of the product, which can be assumed negligible. This information is now added to the revised R1 manuscript.

- L112: Please add a short description and explanation of abbreviations in the table header.

A description of the abbreviations has been added to the header of Table L, as follows: Abbreviations: weighted average cost of capital (WACC), capital expenditures per unit of capacity (capex), annual operational expenditures per unit of capacity (opex_{fix}), variable operational costs (opex_{var}), efficiency (eff.), electricity (el.).

- L118: Please add a comma before “,such as”. The reviewer believes “droughts” instead of “draughts” is meant in this context

Comma is added and the typo is fixed. Thanks for noticing.

- L119: The reviewer believes that the authors mean: maximises the efficiency in using nutrients

True. The wording corrected as suggested.

- L121: the reviewer wonders what “energy system space” means, can the authors please define this term

The following sentence has been added to the main text for clarification: “The production volume, cost, and emissions of e-SCP is largely affected by the availability and cost of renewable electricity, CO₂ DAC, water electrolysers, and other energy conversion technologies, that are traditionally linked to the defossilisation of the energy-industry system. Thus, e-SCP could transform the traditional food production space to an energy system space.”

- L122: The authors believe that “substrate shortage” is a better wording

The wording has been changed to “substrate shortage”.

- L123: Please explain the problem of current fluctuations of the different renewable energies here, and what balancing technologies are needed.

The problem of the variable source of power added to the start of the paragraph.

We need to keep the word count of the Introduction section to a certain limit. Thus, the required balancing technologies were described in the caption of Figure 2. We wish to keep it that way.

- L130: Please always introduce abbreviations: Photovoltaics (PV)

The PV abbreviation is now introduced at the first appearance.

- L144-L147: Please simplify this sentence as, in the opinion of the reviewer, the rationale for the study becomes not clear

The sentence has been simplified.

- L156: “r” is missing, please correct to “learning rate”

Fixed. Thanks for noticing.

- L164: The reviewer thinks that figure 3 (left) is not needed in the main text, because it is not mentioned in the text. The reviewer suggests moving the figure to the supplementary information.

The text has been now adjusted to mention both sub-figures in the text. Please notice that the figure that was originally on the left side has been now moved to the right side and is named ‘b’.

- The reviewer does not understand the axis scale of the blue bars (installed capacity Ref. scenario; additional installed capacity Adv. scenario) as the numbers do not match the right y-axis scale. The change from 2028, 2030 and 2035 is barely readable. The reviewer asks to rescale the axis and enlarge bars.

Thank you for pointing this out. The right scale for the blue bars would have been Mt/timestep, for which the timestep would be 1 year for 2028, 2 years for 2030 and 5 years for 2035, 2040, etc. We have now removed the blue bars from the figure, as they are not mentioned in the text. This would make the figure less complex and obviates the problem with readability of the bars in early years. The underlying data is still available in the Supplementary Table E.

- L169: correct small-scale

Corrected. Thanks for noticing.

- L172: The term “learning by doing” seems wrong here, please change to “learning curve effect”

The term adjusted to “learning curve effect”.

- L178: The reviewer does not understand the term “regional potential” as in this context it seems to depend on “regional availability of renewable energy”

Here we want to address both volume and cost of renewable energy in each region. Thus, we use the wording “regional potential”. Same as the title of the manuscript “Global potential of sustainable single-cell protein...” which addresses both production cost and production volume of e-SCP. We think “Regional availability” would only refer to the regional volume. To avoid any confusion, the respective sentence has been updated as follows:

The cost and production volume of e-SCP is directly affected by the regional potential (generation cost and volume) of renewable electricity.

- L180: please introduce the abbreviation: photovoltaic (PV)

The PV abbreviation is now introduced in the first appearance in the caption of Figure 2, as well as the first appearance in the main text in the last paragraph of the Introduction section.

- L193: Please pay attention to the same terminology in text, subtitle, and figure as it facilitates better comprehension for the reader: cost of electricity supply to e-SCP

Thanks for the comment. The respective subfigure title has been updated from “delivered electricity” to “electricity supply” to match the wording in the text and figure caption.

- L198: Please add (e-protein) after “cost of its protein content” to have a connection to the figure and for easier understanding

Added.

- L204: The reviewer thinks it is better to use the term “Capex” instead of “cost of the SCP core plant”, as the term Capex was also used in the previous chapter

True. The wording changed to “capex”.

- L207: The reviewer asks if the supplementary figures can be labeled uniformly. A different numbering as in the main text is recommended for supplementary figures.

We share the reviewer’s opinion about the labelling format of the supplementary figures in the main manuscript. However, the present format is based on the journal’s guideline. Thus, unfortunately, we think we have to keep the current format.

- L213: Please restructure the whole chapter as the argumentation is not clear to the reviewer, please also refer to Fig. 6 in the text to connect the explanations with the visual information and use the same wording in the text as in the caption of the figure (the reviewer was wondering what are parts of the Power-to steam system L220 in figure 6); explain abbreviations when mentioned for the first time

Thanks for the comment. Indeed, this part is harder to follow. The section has been reworked. The goals are now stated in the first paragraph:

“The objectives of this section are: firstly, to identify the main contributors to the energy consumption and final cost of e-protein; Secondly, to provide an insight on how the sub-units operate and interact in the cost-optimised system, such as the choice of competing technologies and their utilisation rate; Thirdly, to put the scale of the required components, particularly power and hydrogen supply units, into perspective. Such insights help to identify the key factors for further improvements and sensitivity analyses. Moreover, such details would expand the bases for a more detailed analysis and debate on the input data and results of this study by readers.”

The wording in the text and figure caption has been aligned, and sub-figures have been referred to in the text for better clarification. However, the text might still require more effort to follow compared to the other sections. This is because here we do not simply report on results of each sub-figure independently but explain how certain specifications and behaviour of some sub-unit could affect the behaviour of others. This requires bridging among different sub-figures of Figure 6, compared to discussing each one separately. It is a glimpse on how such an energy system could work dynamically. Thus, those with a background in energy systems might find it easier to follow.

- L238: The reviewer does not understand the sentence. Can the authors please explain why higher installed capacities are needed in PV-dominated regions compared to wind-dominated areas

PV plants often have lower capacity factors than wind power plants. Consequently, the electrolyzers powered by PV plants have lower utilisation factor than those powered by wind power plants. As a result, a larger capacity of electrolyzers should be installed in a PV-dominated region compared to the one in a wind-dominated region to provide the same amount of hydrogen: $\text{Generation} = \text{Installed capacity} * \text{Utilisation Factor}$.

Consequently, the over-production of hydrogen at peak hours in a PV-dominated system is larger. This leads to the need for a larger hydrogen compressor system to store the extra production in the

hydrogen storage system for consumption at the times with low direct hourly hydrogen production (please see the hydrogen flow options in Figure 2). Figure 6f visualises the sizing of electrolyzers and hydrogen compressors in PV- and wind-dominated regions. For example, in Chilean Patagonia (CHL-PAT) with excellent wind resources and consequently wind power plants with high utilisation factors, the installed power capacity is the lowest (Figure 6d). The high utilisation factor of wind power in this region also leads to a high utilisation factor or full load hours of electrolyzers (Figure 6c), and consequently lowest installed capacity of electrolyzers and hydrogen compressors (Figure 6f).

Alternatively, batteries could be installed to smooth out the power supply to the electrolyzers in a PV- or wind-dominated region, leading to a smaller electrolyser system with higher utilisation factor. However, batteries are found not to be part of the optimal solution for hydrogen supply to SCP core plant. In other words, balancing the hydrogen supply by oversized electrolyser, and addition of hydrogen compressor and storage system is cheaper than balancing the power input to a smaller electrolyser for hydrogen production at higher utilisation factor.

We have now tried to make the case clearer in the revised manuscript.

- L260: the authors were comparing different costs in different regions before; this information is, however, missing for the non-energetic costs. The reviewer was wondering if these costs are also similar in all areas.

We are not sure if we understand the question correctly. Figure 6b includes the non-energetic cost of sub-units in the 7 nominated sites. From there, one can see that the non-energetic cost of electrolyser and hydrogen balancing package (H₂ compressor and storage) is significantly affected by the region. In the main text, we mention the main non-energetic costs such as SCP plant and the electrolyser. We think providing more details at this section could make the paper too lengthy and turn it to a full technical paper.

- L273: The reviewer thinks Fig. 6 (bottom; left, middle, right) are not mentioned and explained in the main text; please indicate in the text if data is explained by referring to the figure.

This sub-figure is now Fig. 6d. It has been mentioned and explained in the main text.

- L282-L286: Please simplify this sentence and divide the sentence into several short sentences

The sentence is now broken down into three sentences:

“Low-cost e-protein sites are mainly at sunny regions. Thus, we consider the average required power capacities in Australia, Chile, and Germany as a rough mix of expected climates for Power-to-SCP plants. This leads to a global resource mix of 89% to 91% PV power capacity (80% to 82% PV electricity) from 2035 to 2050, respectively for mass deployment of e-SCP plants.”

- L286 and the following: Please exchange “/” by “or” because the authors also use “/” division symbol, making the sentence hard to understand.
True. Exchanged “/” for “and” instead of “or”, as it seems more fitting.

- L292-306: The reviewer suggests moving this paragraph to the previous chapter as it does not fit in the chapter “area demand”

Thanks for pointing out that the mentioned paragraph is not directly represented in the subsection heading. The energy generation and consumption per unit of produced protein are both affected by

the regional cost-optimised system configuration and losses. Figures 7a and 7b (Power-to-SCP overall efficiency) provide the insight on the electricity generation per tonne of protein production. The mentioned paragraph (previously L292-306) explains Figures 7a and 7b and is therefore required in this section. To address the reviewer's rightful concern, we have updated the subsection heading to "Specific regional energy efficiency, area demand, and global production potential of e-protein" to better represent the results presented and discussed in this subsection.

- L298: The reviewer is wondering where to find the data of over 80 MWh/t(protein) in Far East Russia, please include a reference to the data here or include (data not shown)

The regions with electricity generation of over 80 MWh/t,protein are presented in Figures 7a & 7b in areas in dark blue (90°–150° E, 50°–65° N). More clarification and referencing have been added to the text.

- L309: The reviewer wonders what is meant with "here"; please add a reference to the data

"here" was referring to Figures 7 c & d. The main text has been updated with reference to data.

- L378: Please add label next to the y-axis

Done.

- L387: Please add "per year; /a" in the text

This is not the first appearance of the unit "/a" in the text. So, we are not sure if we understand the comment correctly. Our interpretation is that the reviewer wants us to define the full name of the mentioned unit as "per year" in the text. So, we have added that to the first appearance of the unit "/a", which is in the first sentence of the Results section of the revised R1 manuscript. Please let us know if we have not understood the comment correctly.

- L398: Please change: aiming SCP supply at industrial scale

The wording changed.

- L414: correct "progress ratio"

Corrected. Thanks for noticing.

References

- Please correct references towards consistency; Journal pages are often missing.

Thanks for noticing. References have been rechecked and aligned. The missing journal pages have been added.

Reviewer #2 (Remarks to the Author):

In the research paper on single-cell protein (SCP) plants, several remarkable strengths shine through, underlining the significance and rigor of the study. First and foremost, the paper distinguishes itself by offering an incredibly detailed and comprehensive examination of SCP technology. It delves deep into the intricate technical facets, leaving no stone unturned. This meticulous approach ensures that readers gain a profound understanding of SCP plants, making it a valuable resource for researchers and industry professionals alike. One of the paper's standout strengths is its inclusion of a rigorous techno-economic assessment. This adds a layer of credibility and practicality to the research, allowing stakeholders to evaluate the feasibility and economic viability of SCP plant implementation. Such assessments are crucial in bridging the gap between theoretical concepts and real-world applications. Furthermore, the paper's forward-looking perspective is commendable. By projecting SCP plant deployment trends up to the year 2070, it provides insights into the long-term evolution of this technology. This foresight is invaluable, especially in the context of sustainable food production and environmental considerations. The paper distinguishes itself with its complex modeling approach, involving intricate calculations for hourly energy and mass balances within the hybrid PV-wind power-to-SCP supply chain. This level of scientific rigor serves as a robust foundation for evaluating the proposed system's feasibility. Moreover, the paper's scope is vast, covering an array of technologies, from solar and wind power generation to hydrogen production and ammonia synthesis. This allows grasping the intricate interplay of technologies in the SCP plant ecosystem, emphasizing the complexity of the subject matter.

Thank you for your positive response to our research.

However, to enhance accessibility, the paper could benefit from simplifying explanations for readers with varying levels of expertise.

Throughout the revised manuscript, we have implemented the suggested changes by Reviewer #1 and had the manuscript proofread by a native English speaker, with the aim of simplifying the language while maintaining scientific clarity. Such changes can be traced in the 'Track Changes' version of the revised R1 manuscript.

REVIEWER COMMENTS

Reviewer #1 (Remarks to the Author):

The reviewer sincerely appreciates your efforts in responding to the comments and the improvements made in the language of the manuscript. However, the revisions implemented in the main text are insufficient. The manuscript requires substantial improvements in argumentation and structure. Simplification is necessary to make the content accessible to readers with different levels of expertise.

1. The introduction still does not identify the research gap that your study aims to fill. This section needs to clearly articulate the specific problem or gap in the existing research that your study addresses. This is essential to set the context for your work and highlight its significance.

2. Comparisons to other Techno-Economic Analyses (TEAs) are entirely missing, despite it being emphasized in the first revision (for example, a table comparing different 2-3 other TEAs). It is crucial to include these comparisons to validate your numbers. A table summarizing how your results compare to other studies and a simple explanation of why your numbers differ should be added to the main text.

3. The sections “Long-term development and techno-economics data on Solar Foods SCP core plant” and “Energy and cost projection of solid sorbent Direct Air Capture” were added to the supplementary information. Still, the reviewer thinks it should be a substantial and essential part of the main manuscript. The numbers from this study need to be set in relation to other studies. The manuscript still lacks those references to existing literature.

Reviewer #2 (Remarks to the Author):

I commend you on the significant improvements to the manuscript. The revisions have greatly enhanced the clarity and accessibility of the content, simplifying explanations for readers with different expertise levels. This ensures a wider audience, including researchers and industry professionals easily understand the research findings. You've enhanced the study's overall impact by bridging complex technical details with practical understanding.

Two minor corrections would be:

- figure 6 – please define all abbreviations (OCGT, CCGT, etc), revise all figures

- also, please revise the abbreviations consistency i.e. Co₂, H₂, I suggest to use or hydrogen or H₂ for better understandability

Congratulations on your progress, and I look forward to the continued impact of your research in the scientific community.

The authors thank the editor and appreciate the reviewers' valuable comments and suggestions. Each reviewers' comment has been addressed in turn, followed by an appropriate response. All responses are highlighted in blue font for clarity. The corresponding corrections in the revised R2 manuscript can be found in the 'Track Changes' version, provided alongside the regular 'Clean' version.

Reviewer #1 (Remarks to the Author):

The reviewer sincerely appreciates your efforts in responding to the comments and the improvements made in the language of the manuscript. However, the revisions implemented in the main text are insufficient. The manuscript requires substantial improvements in argumentation and structure. Simplification is necessary to make the content accessible to readers with different levels of expertise.

We also sincerely appreciate the reviewer's valuable feedback and thoughtful comments. We made significant efforts to simplify the content in both the original and revised R1 manuscripts. For instance, we deliberately avoided discussing the hourly behaviour of the system components, which would be of interest to researchers with a background in dynamic energy systems but would likely be more difficult for other target audiences of this study to digest. However, finding the balance between simplicity and maintaining the necessary scientific rigor is indeed challenging. This manuscript presents a detailed scientific study on the global optimisation of complex systems with high spatial and temporal resolution. As a result, the number of variables, system dynamics, and results differ significantly from earlier case studies with predefined operational conditions. Therefore, it is essential to provide sufficient detail and transparency regarding the dynamics of the optimised system to properly interpret the results, all of which contribute to a certain level of complexity. Nevertheless, we have made additional efforts to restructure and simplify the content, as reflected in the Track Changes version of the revised R2 manuscript. For example, as explained in the main text, the energy cost of each subcomponent cannot be fully isolated in a dynamic, integrated energy system. However, we have provided a rough estimate of the electricity consumption cost for major subunits. This helps to estimate the total cost of H₂ and CO₂ supply, which is useful for comparing our results with those reported in the literature.

Further simplifications would undermine the scientific rigor, while extended explanations for improved clarity for readers with varying expertise are constrained by the journal's word count limits. Our aim has been to simplify the messaging as much as possible while ensuring that the key aspects of this multidisciplinary topic are not overlooked. Reviewer #2's feedback suggests that we made progress towards this goal in the R1 manuscript. We believe we have further enhanced the clarity and structure in the revised R2 manuscript, thanks to for your additional comments.

1. The introduction still does not identify the research gap that your study aims to fill. This section needs to clearly articulate the specific problem or gap in the existing research that your study addresses. This is essential to set the context for your work and highlight its significance.

We have extensively clarified and elaborated on the research gaps that our study aims to address in the final three paragraphs of the Introduction section of the revised R2 manuscript, prior to Figure 2.

2. Comparisons to other Techno-Economic Analyses (TEAs) are entirely missing, despite it being emphasized in the first revision (for example, a table comparing different 2-3 other TEAs). It is crucial to include these comparisons to validate your numbers. A table summarizing how your results compare to other studies and a simple explanation of why your numbers differ should be added to the main text.

Thank you for highlighting this point. We have now identified five journal articles on TEA of e-SCP, including one published after the initial submission of our original manuscript. Table 1, at

the end of the Results section, summarises the key factors from these TEAs and compares them to our assumptions and results. Additionally, throughout the Results section, we compare our findings with insights from the literature, including techno-economic assumptions, cost of intermediate H₂, CO₂, and the final product, as well as land use impacts. Due to the word count constraints in the main manuscript, a more in-depth analysis of the available literature is provided in Supplementary Section S3.3. Supplementary Figure S2 visualises the key data from Table 1 in the main text. We would like to note that not all the data in Table 1 were directly available from the reviewed literature, as the studies varied in their levels of data disclosure, results presentation, and cost assessment methods. Therefore, to facilitate a meaningful comparison, we made an effort to reconcile these discrepancies and reproduce the necessary input data and results using the levelized cost method.

Since the available literature on cost, energy and mass balance of e-SCP is mainly based on CDW content, we have also provided our result in corresponding units such as €/t_{CDW}, t_{gas}/t_{CDW}, and MWh/t_{CDW}. This approach is preferable as the protein content of CDW in the literature is typically based on broad assumptions rather than the product composition. The units based on protein content are also retained, as they are most suitable for comparison with other protein-rich food and feed sources in the Discussion section. The changes are visible in the 'Track Changes' version of the revised R2 manuscript.

3. The sections “Long-term development and techno-economics data on Solar Foods SCP core plant” and “Energy and cost projection of solid sorbent Direct Air Capture” were added to the supplementary information. Still, the reviewer thinks it should be a substantial and essential part of the main manuscript. The numbers from this study need to be set in relation to other studies. The manuscript still lacks those references to existing literature.

We aimed to include as much content as possible in the main manuscript. However, in accordance with the journal's guidelines, we are limited to 5000 words for the main text, 3000 words for the Methods section, and a maximum of 10 displays (figures or tables combined). We have already exceeded the word count limits. The revised R1 manuscript contained 8 figures, and an additional display (Table 1, comparing our results with available TEAs) has been included in the revised R2 manuscript based on the previous comment. This leaves room for one more display in the main paper. Therefore, Table 2 has been added to the Methods section, summarising data for the reference scenario in the section “Long-term development and techno-economics data on Solar Foods SCP core plant”. Additionally, the Methods section has been expanded with a new subsection, “Energy and mass balance of SCP production”, which provides an overview of the key techno-economic specifications of the SCP core plant and projected trends over time.

Respectfully, we have kept the section “Energy and cost projection of solid sorbent Direct Air Capture” in the Supplementary Information. This section, comprising three displays and approximately 700 words, could not be accommodated within the main manuscript due to space limitations. As the analysis of DAC technology is not the central focus of this study, we believe it would be doable to retain this section in the Supplementary Information. However, we have expanded our explanation in the Results section to further elaborate on the contribution of CO₂ to e-protein production costs and how it compares with the literature. Additionally, we have introduced a new section on the sensitivity analysis of major components, which discusses the impact of long-term energy and cost projections for DAC on e-protein production costs.

Reviewer #2 (Remarks to the Author):

I commend you on the significant improvements to the manuscript. The revisions have greatly enhanced the clarity and accessibility of the content, simplifying explanations for readers with different expertise levels. This ensures a wider audience, including researchers and industry professionals easily understand the research findings. You've enhanced the study's overall impact by bridging complex technical details with practical understanding.

We appreciate your supportive feedback.

Two minor corrections would be:

- figure 6 – please define all abbreviations (OCGT, CCGT, etc), revise all figures

OCGT and CCGT were previously defined in the caption of Figure 2. For added clarity, they are now also defined in the caption of Figure 6.

- also, please revise the abbreviations consistency i.e. Co₂, H₂, I suggest to use oxygen or hydrogen or H₂ for better understandability

True. We reviewed the options and opted to use H₂, O₂, and CO₂ instead of their full names throughout the manuscript. The only exception is the abstract, where good journal practices recommend avoiding both abbreviations and chemical formulas.

Congratulations on your progress, and I look forward to the continued impact of your research in the scientific community.

Many thanks for your very positive feedback.

REVIEWERS' COMMENTS

Reviewer #1 (Remarks to the Author):

I have reviewed the revised manuscript submitted to NatComm and confirm that the authors have addressed the changes I requested. The authors have revised the introduction and clarified the research gap their study addresses. They have incorporated a comparative analysis of their results with other techno-economic analyses (TEA) on microbial protein production and summarized it in a clear Table 1. This strengthens the context and relevance of their findings. I am satisfied with the revisions and believe the manuscript has been significantly improved.